# LLaVAction: Evaluating and Training Multi-Modal Large Language Models for Action Understanding

**Haozhe Qi**[*], **Shaokai Ye**[*], **Alexander Mathis**[**✉], **Mackenzie Weygandt Mathis**[**✉]

École Polytechnique Fédérale de Lausanne (EPFL), Lausanne

✉ mackenzie.mathis@epfl.ch, alexander.mathis@epfl.ch [*]Co-first, [**]Co-senior

## Abstract

Understanding human behavior requires measuring behavioral actions. Due to its complexity, behavior is best mapped onto a rich, semantic structure such as language. Emerging multimodal large language models (MLLMs) are promising candidates, but their fine-grained action understanding ability has not been fully examined. In this work, we reformulate EPIC-KITCHENS-100, one of the largest and most challenging egocentric action recognition datasets, into a MLLM benchmark (EPIC-KITCHENS-100-MQA). We demonstrate that when we sample difficult answers based on specialist models as distractors, leading MLLMs struggle to recognize the correct actions. How can we increase the performance of MLLMs? We curated a supervised finetuning dataset that includes 'hard' action recognition, temporal detection, captioning, and free-form question answering to improve models' diverse action understanding capabilities. We introduce a new model called LLaVAction that adds an action token to boost models' attention on visual tokens and a two-stage pipeline to obtain structured actions. LLaVAction greatly improves the MLLMs' ability of action understanding, achieving strong improvements on both MLLM benchmarks (21 points in accuracy over GPT-4o on EPIC-KITCHENS-100-MQA) and established action recognition benchmarks, suggesting that our methods prepare MLLMs to be a promising path forward for complex action tasks. Code, data, the benchmark, and models are available at https://github.com/AdaptiveMotorControlLab/LLaVAction.

## 1 Introduction

Understanding human behavior is a complex challenge requiring multiple skills such as visual perception, knowledge about the world and reasoning capabilities. Current State-of-the-Art (SOTA) methods in action understanding tasks (Chalk et al., 2024; Liu et al., 2025; Shi et al., 2023) typically rely on visual foundation models to imbue those kind of priors (Radford et al., 2021; Wang et al., 2022; 2023). However, they rely heavily on dataset-specific target heads and have limited language understanding ability, constraining their performance and especially generalizability. Recently, Multi-modal Large Language Models (MLLMs) (Zhang et al., 2024c; Li et al., 2024a; et al., 2024; Wang et al., 2024) have shown great potential for learning language priors to help understand visual content, making them promising alternatives.

MLLMs take visual and text information as inputs and can directly output text. For training and evaluating MLLMs on action understanding tasks, existing datasets (Kay et al., 2017; Caba Heilbron et al., 2015) are converted into free text (either video caption or question-and-answer [QA] formats), thus creating new datasets (Liu et al., 2024a; Li et al., 2024d) and benchmarks (Yu et al., 2019; Li et al., 2024c). Those free-text formats offer great flexibility and generalization across datasets, but also introduce limitations in model learning, evaluation, and application perspectives. For the model learning, directly predicting the action name or choosing from some randomly selected candidates (Figure 1) prevents the model from learning the full action distributions and contrasting fine-grained actions (Xiao et al., 2021) explicitly. For the model evaluation, free text output makes MLLMs unable to directly compare with previous action task specialized models (Ramachandran et al., 2025).

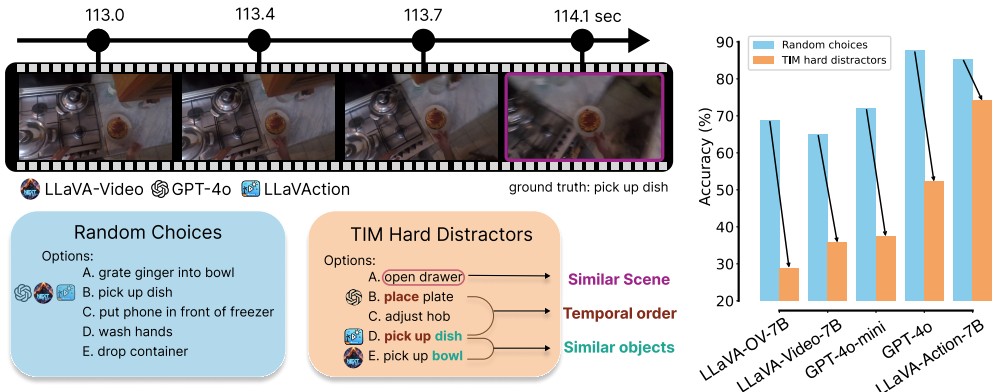

Figure 1: **LLaVAction-7B.** Left: Qualitative inspection of distractors. We show an example clip with labels from random choices (which empirically is easy to solve), vs. our proposed harder benchmark with action labels generated by a SOTA specialist (TIM (Chalk et al., 2024)). Our hard example mining strategy can automatically explore challenges such as temporal order and similar objects that are curated in other benchmarks. Right: While GPT-4o is strong when identifying correct answers among few random choices due to the large number of possible actions, it suffers in the harder benchmarking regime, and our method, LLaVAction outperforms GPT-4o.

For example, EPIC-KITCHENS-100 (Damen et al., 2022), which is one of the largest and most challenging action datasets, has around four thousand actions. MLLMs may not always predict an action that has an exact match in those action types, as we cannot put all the action types inside MLLMs' context prompt to let MLLMs select. This further limits the applications that require structured actions (e.g., behavior analysis with ethograms (Renner, 2018)).

To address these issues, we take inspiration from the hard example mining literature (Shrivastava et al., 2016; Madry, 2017) to improve the learning and evaluation of MLLMs. Specifically, for evaluation, we reformulate EPIC-KITCHENS-100 (Damen et al., 2022) into a video multiple-choice question & answer (MQA) task with the correct ground truth action and four difficult incorrect actions, which we call EPIC-KITCHENS-100-MQA. Incorrect choices are filtered by SOTA action recognition models (Chalk et al., 2024; Zhao & Krähenbühl, 2023) instead of humans or closed-sourced LLMs and MLLMs. This specialized model-based hard example mining reveals substantial drops in performance for existing MLLMs, including GPT-4o (Figure 1) and thus offers an efficient and challenging framework for evaluating MLLMs' action recognition abilities. We note that the hard example mining approach automatically picks distractors that pose challenges such as temporal order, which were purposefully curated in other benchmarks (Cai et al., 2024; Li et al., 2024c).

To improve MLLMs' fine-grained action understanding, we proposed an action-related MLLM data transformation regime and curated a training dataset that encompasses various aspects of action understanding, such as hard action recognition, detailed captioning, free-form question answering, temporal detection and prior action association. With the training dataset, we propose LLaVAction models. We introduce an action token designed to improve the model's visual information utilization and a two-stage pipeline to output structured actions and fairly compare with other action recognition models. These model designs could be naturally extended to different foundational MLLMs (e.g., Liu et al. (2024a); Zhu et al. (2025)). LLaVAction obtains SOTA performance on four action recognition datasets (EPIC-KITCHENS-100 (Damen et al., 2022), EPFL-Smart-Kitchen-30 (Bonnetto et al., 2025), MECCANO (Ragusa et al., 2021) and Animal Kingdom (Ng et al., 2022)) and shows strong generalization ability in comparison to previous SOTA models. LLaVAction outperforms GPT-4o on EPIC-KITCHENS-100-MQA and achieves consistent improvements on ten video MLLM benchmarks that require very different action understanding abilities and are either in a caption, open-ended, or multi-choice format.

## 2 RELATED WORKS

**Multi-modal large language models.** Multi-modal large language models (MLLMs) are promising generalists (Li et al., 2024b). Early multi-modal models (Tsimpoukelli et al., 2021) mostly

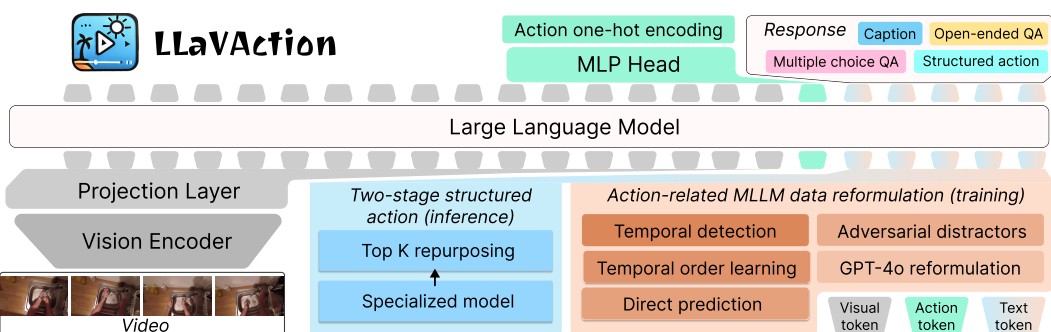

Figure 2: **LLaVAction pipeline.** Trained with our action-related MLLM reformulated data, LLaVAction outputs captions, action tokens and open-ended and multi-choice QAs. Our two-stage pipeline further enables LLaVAction to output structured action.

performed few tasks or relied on few-shot learning for task generalization. After the large success of Large Language Models (Achiam et al., 2023), multi-modal models appeared that can supplement text with other modalities (Han et al., 2024). Among them, video MLLMs (Li et al., 2024a; Zhang et al., 2024c) promise robust and scalable solutions to understand and process video data. Our work falls into this direction, aiming at improving MLLMs' action understanding. Action understanding is one of the fundamental abilities in video understanding and has been explored by several recent works with different specific focuses, such as InsTALL (Nguyen et al., 2025) mainly focusing on procedural action planning/prediction, HAIC (Wang et al., 2025a) mainly focusing on detailed action captions and MotionLLM (Chen et al., 2024b) mainly focusing on human motion understanding. Instead, our work cares more about fine-grained action contrastiveness.

**MLLM datasets and benchmarks.** Significant efforts have been made to improve training (Liu et al., 2024a; Li et al., 2024d) and benchmarking (Yue et al., 2024; Liu et al., 2024c) for MLLMs. Based on the question type, they can be classified as video caption, open-ended question answering and multi-choice question answering (MQA) types. MQA format gains more popularity, especially for benchmarks (Xiao et al., 2021; Fu et al., 2024), since it has no need to use another LLM/M-LLM to evaluate the model outputs and our EPIC-KITCHENS-100-MQA benchmark falls into this direction. In comparison to the existing MQA benchmarks whose choices are either constructed by humans (Yu et al., 2019; Fu et al., 2024) or by closed-source MLLMs (Maaz et al., 2023; Ye et al., 2024a), our benchmark uses action recognition models to efficiently find hard distractors, which is more efficient compared to human generation and is not be limted by closed-source MLLMs' performance.

**Action recognition.** Action recognition requires models to predict the action class for a trimmed segment (Shahroudy et al., 2016; Damen et al., 2022) and is a fundamental task in video understanding (Feichtenhofer et al., 2019; Tong et al., 2022). Over the years, many methods have been proposed, yet suffer from fast camera movement, long-term temporal relations, and open vocabulary ability (Damen et al., 2022; Grauman et al., 2022). We focus on MLLMs enhanced with video instruction-tuning (Zhang et al., 2024c) to address those challenges.

# 3 METHODS

We introduce the EPIC-KITCHENS-100-MQA benchmark (Section 3.1) and a novel MLLM data reformulation paradigm (Section 3.2), followed by the LLaVAction model designs (Section 3.3).

## 3.1 HARD EXAMPLE MINING FOR MLLM EVALUATION

Existing MLLMs have shown the ability to understand video content including actions. However, whether MLLMs are good at contrasting fine-grained actions is not clear. Researchers have developed benchmarks to focus on certain aspects of fine-grained actions, such as temporal order (Liu et al., 2024d), which are generated either by human effort or closed-source MLLMs. In this work, we propose to leverage SOTA action recognition models with hard example mining to construct a new benchmark named EPIC-KITCHENS-100-MQA, which is more efficient compared to human gen-

eration and is not limited by closed-source MLLMs' performance. More importantly, the proposed hard example mining paradigm can also help MLLMs to enhance fine-grained action understanding (Section 3.2) and enable the fair comparisons with specialized models (Section 3.3).

We use EPIC-KITCHENS-100 (Damen et al., 2022) as the data source for our benchmark for the following reasons. Firstly, EPIC-KITCHENS-100 boasts fine-grained action at scale (90K action segments comprising 100h, 4k action types in 100 hours). Secondly, despite numerous models being developed for this dataset, benchmark performance on tasks such as action recognition and segmentation remains far from saturated. Thirdly, the benchmark proves opportunities to compare against specialized models. Importantly, our hard example mining strategy is generalizable and can be applied to any other action understanding dataset. (Section 4.3).

We constructed EPIC-KITCHENS-100-MQA with hard example mining as follows: Let $\mathcal{V} = \{v_1, v_2, ..., v_N\}$ denote the set of video clips. Let $\mathcal{N} = \{n_1, n_2, ..., n_N\}$, $\mathcal{A} = \{a_1, a_2, ..., a_N | a_i \in \mathcal{C}\}$ be their corresponding clip narrations and action labels separately, where $\mathcal{C}$ represents the set of action classes. For each data sample $i$, we formulate the MQA task as:

$$f : (v_i, \mathcal{Q}, \mathcal{O}_i) \mapsto [p_1, p_2, \ldots, p_K] , \text{where} \quad \sum_{k=1}^{K} p_k = 1, \quad p_k \in [0, 1] \tag{1}$$

where $v_i$ is the input sample (e.g., video clip), $p_i$ is the probability of picking the i-th option in the MQA as the answer, $\mathcal{Q}$ is the space of possible questions, $\mathcal{O}_i = \{n_i, \mathcal{D}_i\}$ represents the set of $K$ answer options, $n_i$ is the correct narration, and $\mathcal{D}_i$ represents $K - 1$ sampled distractors. These can be sampled randomly from narrations in other action classes:

$$\mathcal{D}_i^r = \text{Uniform}(\{n_j \in \mathcal{N} \mid c_j \in \mathcal{C} \setminus \{a_i\}\}) \tag{2}$$

However, random sampling $\mathcal{D}_i^r$ likely contains trivially wrong answers (Figure 1). We utilize action recognition models $g : \mathcal{V} \to (0, 1)^{|\mathcal{C}|}$ to find distractors. For video a specific $v_i$, we obtain the top $K - 1$ predicted classes: $\mathcal{C}i = \text{Top}_{K-1}(g(v_i) \setminus \{a_i\})$. The distractor sampling becomes:

$$\mathcal{D}_i^m = \text{Uniform}(\{n_j \in \mathcal{N} \mid c_j \in \mathcal{C}_i\}) \tag{3}$$

The complete set of answers is formed as $\mathcal{O}_i^r = \{n_i\} \cup \mathcal{D}_i^r$ for random sampling or $\mathcal{O}_i^m = \{n_i\} \cup \mathcal{D}_i^m$ for model-based sampling. We used $K = 5$ for our benchmark. Moreover, we use the action narrations $\mathcal{N}$ instead of action labels $\mathcal{C}$ to build the choices to avoid implausible texts and confusions (more details are in Appendix F.1) We compared the two sampling strategies. We chose two leading action recognition methods on EPIC-KITCHENS-100, namely, AVION (Zhao & Krähenbühl, 2023) and TIM (Chalk et al., 2024). The results indicate that the TIM method consistently produced more challenging distractors for the evaluated MLLMs (Table 1 and qualitative examples Appendix Figure 6). Consequently, we fixed $g$ with TIM for the EPIC-KITCHENS-100-MQA benchmark. We found that all tested models have a huge performance drop in comparison to the easy setting, which illustrates that MLLMs struggle with fine-grained action recognition when tested with visually or semantically similar actions.

## 3.2 ACTION UNDERSTANDING FROM MULTIPLE PERSPECTIVES

When training MLLMs on action understanding datasets (Kay et al., 2017), researchers usually re-formulate those datasets to MLLM-compatible tasks (video caption or question & answer (QA)), either by directly outputting the original action annotation or reformulating the annotations with closed-source MLLMs. Although this reformulation provides flexibility and generalization across video data even beyond action understanding, fine-grained action differences are not fully explored and hence

| Choice Selection | Random 5 (Easy) | Avion-Top 5 (Medium) | TIM-Top 5 (Hard) |
|---|---|---|---|
| GPT-4o-mini (07-18) | 72.0 | 44.2 | 37.4 |
| GPT-4o (08-06) | 87.6 | 56.7 | 52.2 |
| LLaVA-OV-0.5B | 59.3 | 37.1 | 32.0 |
| LLaVA-OV-7B | 68.8 | 33.6 | 28.9 |
| LLaVA-Video-7B | 65.0 | 40.0 | 35.7 |

Table 1: **Comparison between different sources of distractors.** Models were evaluated on either random, AVION or TIM-generated distractors. The values are reported as percent accuracy.

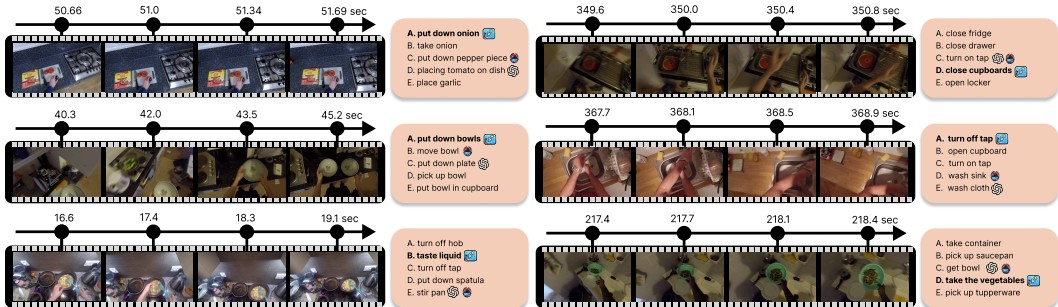

Figure 3: **Qualitative results.** LLaVAction-7B consistently outperforms GPT-4o and LLaVA-Video-7B when tested on hard distractors. Bold option denotes ground truth, and the icons denote the selection of the models. See also Appendix F.

cause a performance drop on EPIC-KITHCENS-100-MQA. To address this, we expand the previous reformulation regime to encompasses various aspects of action understanding.

**Adversarial distractors for fine-grained action contrasting.** Thanks to the effectiveness of our hard example mining paradigm, we can also train MLLMs with 'adversarial' distractors that were generated by action recognition models. Since the EPIC-KITCHENS-100-MQA benchmark is based on TIM's predictions, training MLLMs to pick the right answers using distractors generated by TIM could lead to over-fitting to TIM's error distributions, resulting in an independent and identically distributed (IID) setting. To avoid methods from obtaining better performance by simply overfitting on TIM's choice combinations, we used predictions from AVION to reformulate the EPIC-KITCHENS-100 training set. This provides us with an out-of-distribution (OOD) evaluation on EPIC-KITCHENS-100-MQA (more OOD supports in Appendix D). Moreover, we also present the results derived from training with distractors generated by TIM, which is the IID setting and yields the strongest results in EPIC-KITCHENS-100-MQA (Table 2).

**Temporal detection for action boundary learning.** Many actions (such as putting something) have clear initiation and conclusion and cannot be explicitly learned by the previous reformulation regime. Therefore, we instruct the model to predict the start and end timestamps of an action sample based on its randomly padded video clip. Specifically, given a video segment $v_i$ with start timestamp $s_i$ and end timestamp $e_i$, we introduce a fixed temporal padding $\delta = 3$ seconds distributed between start and end. Let $\alpha \sim \text{Uniform}(0, 1)$ be the proportion of padding allocated to the start:

$$\hat{s}_i = s_i - \alpha\delta, \hat{e}_i = e_i + (1 - \alpha)\delta \tag{4}$$

We therefore use the new timestamps $\hat{s}_i$ and $\hat{e}_i$ to obtain the padded video segment $\hat{v}_i$. During training, our LLaVAction model takes $\hat{v}_i$ as input and predicts the start and end times as strings (e.g., "3.20", "1.20") corresponding to the true start and end times of the action in the padded video (prompts are detailed in the Appendix C.3).

**Temporal order learning with prior actions.** Actions exhibit a certain natural continuity. This temporal aspect of actions can improve predictability. Therefore, we want to leverage prior actions and learn: $\theta^* = \arg\max_\theta \sum_{t=n+1}^T \log P_\theta(a_t \mid a_{t-1}, \ldots, a_{t-n})$, where $\theta^*$ is the optimal set of model parameters we are trying to find, $a_t$ is the current action at time t, $a_{t-1}, \ldots, a_{t-n}$ represents the sequence of $n$ previous actions. We set $n = 2$. We modulate this with additional visual instructions to provide prior action information (prompts are detailed in the Appendix C.5). During training, 30% of the MQAs are provided with prior action information as additional input. During evaluation (Table 3), we can either give the model no prior actions or give the model's own predictions of the previous $n$ actions to formulate as sequential action prediction (SAP).

**Direct prediction.** Following common practice, we also let the model to directly predict the action descriptions for a given video. The prompt is available in the Appendix C.4.

**General video understanding with closed-source MLLM-based reformulation.** Although we mainly focus on improving MLLMs' fine-grained action understanding, we do not want to weaken MLLMs' general video understanding. Therefore, we follow previous practice to let the model give a general description of the video or answer video-related questions, where the annotations are

obtained from closed-source MLLM GPT-4o. More details are in the Appendix B. As discussed above, we argue that this reformulation alone would not help to gain better fine-grained action understanding. Since GPT-4o itself struggles with our empirically hard distractors, using annotations from GPT-4o alone harms the performance on EPIC-KITCHENS-100-MQA (Table 1).

## 3.3 LLaVAction models for better action understanding

With the proposed new MLLM data reformulation paradigm, we propose LLaVAction, which is better at fine-grained action understanding. LLaVAction is developed based on LLaVA series (Li et al., 2024a; Zhang et al., 2024c). We further design a learnable action token to enhance the visual information utilization and a two-stage pipeline to output structured action so that we can compare LLaVAction fairly with other SOTA action recognition models (Figure 2).

**Enhancing visual information utilization with action token.** Most MLLMs rely only on the language prediction of the next token to train the model and extract information from visual tokens (Liu et al., 2024a). However, recent findings suggest that this training strategy decreases the importance of vision tokens in late layers of MLLMs (Zhang et al., 2025a; Liu et al., 2024b). Therefore, we designed an intermediate supervision of visual tokens. Specifically, we added a learnable action token into the input tokens, which is analogically similar to the CLS token in the VIT model (Dosovitskiy, 2020) that has been shown the ability in grasping image content (Wang et al., 2025b) The order of the input tokens was system text tokens, visual tokens, learnable action token and instruction text tokens; through causal attentions of the LLM backbone this enables the action token to integrate action information from visual tokens and then contribute to the subsequent language tasks. Let's denote the hidden states at the final layer of the MLLM as:

$$\left\langle H_1^q, \cdots, H_k^q, H_1^v, \cdots, H_{l_v}^v, h^a, H_{k+1}^q, \cdots, H_{l_q}^q \right\rangle \tag{5}$$

$H_i^v \in \mathbb{R}^d$ are the hidden states corresponding to visual tokens, $h^a \in \mathbb{R}^d$ denotes the hidden state of the learnable action token, $H_i^q \in \mathbb{R}^d$ denotes the text tokens, $d$ denotes the hidden dimension of the LLM. $l_q$ and $l_v$ denote the length of text tokens and length of visual tokens respectively. We apply three classification heads on top of the hidden state $h^a$ to predict nouns, verbs, and actions separately and use cross-entropy loss to train the classifiers, in the belief that the classification training could guide the action token to learn better extract action-related visual information. Note that our action token only serves as an additional learning objective and is not used to give the final prediction. During inference, or when training with tasks that have no clear action labels (e.g., video captioning), we can simply compute the text generation loss.

**Two-stage structured action prediction pipeline.** MLLMs directly output free texts, which may be hard to find an exact match with the action labels in the dataset. One could put all the possible actions in the question prompt and let MLLMs choose, but this will become infeasible when the number of action types increases (e.g., around 4k action types in EPIC-KITCHENS-100). This limitation prevents MLLMs from fairly comparing with action recognition models and constrains MLLM from applying to applications that require structured actions. To this end, LLaVAction designs a two-stage pipeline when it needs to output structured actions. Similar to hard example mining (Section 3.1), we use action recognition

| Methods | 8 f | 16 f |
|---|---|---|
| zero-shot GPT-4o | 52.2 | N/A |
| zero-shot GPT-4o-mini | 37.4 | N/A |
| zero-shot LLaVA-Video-7B | 35.7 | 34.8 |
| zero-shot LLaVA-OV-7B | 28.9 | 30.5 |
| zero-shot LLaVA-OV-0.5B | 32.0 | 31.6 |
| LLaVAction: LLaVA-Video-7B | 71.7 | **73.4** |
| LLaVAction: LLaVA-OV-7B | 71.3 | 72.3 |
| LLaVAction: LLaVA-OV-0.5B | 64.8 | 65.4 |

Table 2: **Comparison on EPIC-KITCHENS-100-MQA.** Columns represent the number of frames used for testing. Percent accuracy is shown.

models' predictions to filter out easy actions. The difference is that we directly take the top K confident actions without adding the Ground Truth (GT) action. The two-stage pipeline will lower the upper-bound performance MLLM can obtain since the GT action may not be in the top K prediction. However, the K value can control the trade-off between the upper-bound performance and the number of actions the MLLM needs to contrast. Note that our two-stage pipeline is only needed for datasets and applications that require structured actions, and is only applied at the inference stage. For the open-vocabulary tasks, LLaVAction can directly take the video and questions and give an open-ended answer. For training and benchmark construction, we always include the GT action into the choices to avoid misalignment.

## 4 EXPERIMENTS

We conducted evaluations for the LLaVAction models on a wide range of benchmarks to show the models' strong action understanding ability. We used LMMs-Eval (Zhang et al., 2024a) to evaluate LLaVAction on different MLLM benchmarks. We used eight frames for distractor experiments (Table 1), 8 or 16 frames for our main results on EPIC-KITCHENS-100-MQA (Table 2) and 16 frames for ablation studies (Table 11), so that we can limit the cost of calling closed-sourced models during comparison. Meanwhile, we use 32 frames for EPIC-KITCHENS-100 (Table 4) and 64 frames for the model reported in MLLM benchmarks (Table 7).

### 4.1 IMPLEMENTATION AND TRAINING DETAILS

We trained LLaVAction with three open-source baseline variants, LLaVA-Video-7B, LLaVA-OV-7B and LLaVA-OV-0.5B. With the proposed new MLLM data reformulation paradigm, we re-annotated EPIC-KITCHENS-100's training data, in total contributing to 530K annotated video-language pairs to train the LLaVAction models. The 7B and 0.5B models were trained for 12 and 11 hours on 32 GH200 GPUs, respectively. Across all experiments and all baseline models, we set gradient accumulation to 2, batch size to 64 and total epochs to 2. Following LLaVA-Video, the MLP connector, LLM backbone (Qwen2-0.5B,7B (Wang et al., 2024)) and the visual encoder (SigLIP-384 (Zhai et al., 2023)) were trained. The learning rate was 2e-6 for the vision encoder and 1e-5 for the rest (see Zhang et al. (2024c) for additional details). Training on EPIC-KITCHENS-100 data alone might result in over-fitting. Therefore, we use data replay (Li et al., 2024a) to aid in generalization of the model. Thus, we mix with the training data of LLaVA-Video, namely LLaVA-Video-178K.

### 4.2 RESULTS ON EPIC-KITCHENS-100-MQA

EPIC-KITCHENS-100-MQA contains hard distracting choices and is excellent to evaluate MLLMs' fine grained action understanding. We report results for the MLLMs comparison (Table 2), LLaVAction additive ablations (Table 3), and leave-one-out ablations (Appendix E.2) on EPIC-KITCHENS-100-MQA. We start with MLLM comparisons (Table 2). LLaVAction models perform much better than baselines and also obtain a 21-point improvement over GPT-4o (running GPT-4o beyond 8 frames is cost-prohibitive). Next, we verified LLaVAction's improvements do not just come from adding in-domain data by additive ablations (Table 3). Since both our MLLM data reformulation paradigm (Section 3.2) and LLaVAction model design (Section 3.3) contribute to the final performance, we ablate them together. We can see training with adversarial distractors (AVION) results in the largest improvement (9.4 points). Addition of the action token gives the second most improvements (3.9 points improvement). Meanwhile, we note that simply fine-tuning LLaVA-Video-7B with the previous MLLM

| LLaVA-Video-7B $\Rightarrow$ LLaVAction-7B | |
|---|---|
| **OOD Setting**: | |
| Zero-shot | 34.8 |
| + GPT-4o-based reformulation | 21.9 |
| + Random distractors | 55.0 |
| Adversarial distractors (AVION) | 64.4 |
| + Temporal Detection | 65.2 |
| + Action token | 69.1 |
| + GPT-4o-based reformulation | 71.5 |
| + Direct Prediction | 73.6 |
| + Temporal order learning | 73.4 |
| + Temporal order learning w/ SAP | **74.1** |
| **IID Setting**: | |
| + Adversarial distractors (TIM) | 76.3 |
| + Adversarial distractors (TIM) w/ SAP | 77.0 |

Table 3: **LLaVAction additive ablations on EPIC-KITCHENS-100-MQA**. Techniques are gradually added to achieve the final model. SAP denotes sequential action prediction during inference. Percent accuracy is shown.

data reformulation paradigm (GPT-4o-based reformulation) results in a performance degradation (35.7 to 21.9). Based on the fact that it gives a meager 2.4 performance boost when we combine it with MQA task using AVION distractors, we believe this is a sign of catastrophic forgetting of MQA capability. Furthermore, directly predicting the action (direct prediction) and using contextual prior actions (temporal order learning w/ SAP) result in 2.1 and 0.5 points improvements, respectively. In summary, the combination of our MLLM data reformulation paradigm and model design greatly improves the performance of the base LLaVA-Video-7B model from 34.8 to 74.1 accuracy in the OOD setting and to 77.0 in the IID setting.

### 4.3 RESULTS ON ACTION RECOGNITION BENCHMARKS

With our two-stage structured action prediction pipeline (Section 3.3), LLaVAction can be fairly compared with other action recognition models (Table 4). To assess the generalization of LLaVAc-

tion, we tested on three datasets, which were carefully selected to exclude data used in pretraining while covering different domains.

**LLaVAction achieves SOTA on EPIC-KITCHENS-100.** Following common practice, we report the performance on EPIC-KITCHENS-100's validation set. We report performance in two settings ('w/ action label' and 'w/ action narration'), which differ in the candidate choice generation. For the 'w/ action label' setting, we directly concatenated verb and noun action classes to obtain choices, which could produce implausible text. For example, the noun class for coffee maker represents 'maker:coffee' in the noun definition. Meanwhile, 'pour into' is simplified as 'pour' in the verb definition, which could generate implausible text such as 'pour pot' that should be 'pour into pot'. Based on those observations, we also report the 'w/ action narration' setting where we

| Methods | Acc. |
|---|---|
| IPL (Wang et al., 2021) | 41.0 |
| LaViLa (Zhao et al., 2022) | 51.0 |
| TAdaFormer-L/14 (Huang et al., 2023) | 51.8 |
| LVMAE (Gundavarapu et al., 2024) | 52.1 |
| M&M (Xiong et al., 2022) | 53.6 |
| AVION (Zhao & Krähenbühl, 2023) | 54.4 |
| TIM (Chalk et al., 2024) | 56.4 |
| Ours, LLaVAction-7B w/ action label | **58.3** |
| Ours, LLaVAction-7B w/ action narration | **63.2** |

Table 4: **Action recognition on EPIC-KITCHENS-100.** Top-1 accuracy on action classification. For specific verb-noun performance see Figure 8.

used the action narration of the corresponding video clip (more details in Appendix F.1). We empirically observe that we get better results scaling top-K from 5 to 20. Therefore, we train and evaluate LLaVAction with TIM's top 20 action predictions. LLaVAction achieves SOTA on EPIC-KITCHENS-100 under both settings.

**LLaVAction generalizes well to other datasets.** We tested LLaVAction on two recent action recognition benchmarks – one testing generalization for a different cooking dataset (EPFL-Smart-Kitchen-30, (Bonnetto et al., 2025)) and for a different domain (tool assembly, Meccano, (Ragusa et al., 2021)). EPFL-Smart-Kitchen-30 has 30 verbs and 46 nouns in common with EPIC-KITCHENS-100, which enables us also to compare with specialized models (such as AVION) for zero-shot generalization of second-stage models.

| Methods | Acc. | Head Acc. | Tail Acc. |
|---|---|---|---|
| *Zero-shot generalization of the second-stage model* | | | |
| AVION (Zhao & Krähenbühl, 2023) | 19.3 | 21.2 | 8.6 |
| LLaVA-Video-7B (Zhang et al., 2024c) | 22.5 | 22.9 | 18.8 |
| Ours, LLaVAction-7B | **36.2** | **38.1** | **24.6** |
| *Trained model* | | | |
| VideoMAE (Tong et al., 2022) | 37.5 | 41.1 | 16.6 |
| Multi-modal VideoMAE (Bonnetto et al., 2025) | 40.0 | 43.6 | 19.4 |
| Ours, LLaVAction-7B | **46.6** | **49.7** | **27.0** |

Table 5: **Action recognition on EPFL-Smart-Kitchen-30.** LLaVAction outperforms prior methods in the zero-shot and finetuned setting.

We used the multi-modal VideoMAE's top 5 predictions in EPFL-Smart-Kitchen-30 to generate the MQAs for LLaVAction-7B and LLaVA-Video-7B. To fairly compare with the specialized model AVION, we also use those top 5 predictions to filter AVION's predicted action logits. LLaVAction obtained better zero-shot accuracy than AVION and LLaVA-Video (Table 5). Most importantly, the zero-shot LLaVAction even obtained similar overall performance to the trained VideoMAE model and better tail action accuracy (Tail Acc.). When finetuning LLaVAction obtained SOTA performance (Table 5).

On Meccano we used SlowFast (Feichtenhofer et al., 2019) to generate hard distractors and finetuned LLaVAction. Even when trained for only one epoch, LLaVAction obtained 51.7 top-1 accuracy, beating SlowFast's 42.8.

Our training pipeline and model design is not limited to any particular domains or base MLLMs. To further support that, we tested our method on a very different domain, animal fine-grained behavior understanding. Specifically, we tested on the Animal Kingdom dataset (Ng et al., 2022), which has 140 fine-grained actions and at most 12 actions can happen at the same time (i.e., multi-classification task). We adapted both LLaVA-Video-7B and InternVL3-8B (Zhu et al., 2025) and trained with hard examples generated from Video-MAE (Mamooler et al., 2025) to serve as our methods;

| Methods | Jaccard Acc. |
|---|---|
| VideoMAE (Mamooler et al., 2025) | 53.1 |
| LLaVA-Video-7B (Zero-shot) | 30.5 |
| LLaVA-Video-7B (Random choice) | 46.7 |
| Ours, LLaVA-Video-7B | 61.0 |
| InternVL3-8B(Zero-shot) | 28.1 |
| InternVL3-8B(Random choice) | 43.9 |
| Ours. InternVL3-8B | 58.7 |

Table 6: **Action recognition on Animal Kingdom.**

for two base models (LLaVA-Video-7B and InternVL3-8B) we compared to zero-shot and training with randomly generated options. Our methods, either using LLaVA-Video-7B or InternVL3-8B

| | Caption | | Open-ended Q&A | | | | | Multi-choice Q&A | | | | | |
| --- | --- | --- | --- | --- | --- | --- | --- | --- | --- | --- | --- | --- | --- |
| | VDC (Chai et al., 2024) | VideoDC (Chen et al., 2024a) | VideoEval-Pro (Ma et al., 2025) | ActNet-QA (Yu et al., 2019) | VideoChatGPT (Maaz et al., 2023) | CVRR (Khattak et al., 2025) | TempCompass (Liu et al., 2024d) | EgoSchema (Mangalam et al., 2023) | MVBench (Li et al., 2024c) | VideoMME wo/w-subs (Fu et al., 2025) | LongVideoBench (Wu et al., 2024) | NextQA (Xiao et al., 2021) | PerceptionTest (Patraucean et al., 2023) |
| **Closed-source models** | | | | | | | | | | | | | |
| GPT-4V (Achiam et al., 2023) | - | - | - | - | - | 70.8 | - | - | 43.5 | 59.9/63.3 | 61.3 | - | - |
| GPT-4o (et al., 2024) | - | - | 34.2 | - | - | - | - | - | - | 71.9/77.2 | 66.7 | - | - |
| Gemini-1.5-Flash (Team et al., 2023) | - | - | 35.1 | - | - | - | - | 65.7 | - | 70.3/75.0 | 61.6 | - | - |
| Gemini-1.5-Pro (Team et al., 2023) | 41.7 | - | 39.3 | - | - | - | - | 72.2 | - | 75.0/81.3 | 64.0 | - | - |
| **Open-source models** | | | | | | | | | | | | | |
| LongVA-7B (Zhang et al., 2024b) | 34.5 | - | 16.5 | 50.0 | - | - | - | - | - | 52.6/54.3 | - | 68.3 | - |
| mPLUG-Owl3 (Ye et al., 2024b) | 38.9 | - | - | - | - | - | 34.4 | - | 54.5 | 59.3/68.1 | 52.1 | 78.6 | - |
| VideoChat2-7B (Li et al., 2024c) | 36.5 | - | - | 49.1 | - | 25.8 | 38.5 | - | 60.4 | 42.3/54.6 | - | 78.6 | - |
| VideoLLaMA2-7B (Cheng et al., 2024) | - | - | - | 53.0 | - | 21.6 | 32.2 | 51.7 | 54.6 | 47.9/50.3 | - | - | 51.4 |
| LLaVA-OV-7B (Li et al., 2024a) | 38.8 | - | - | 56.6 | - | - | - | 60.1 | 56.7 | 58.2/61.5 | 56.5 | 79.4 | 57.1 |
| LLaVA-Video-7B (Zhang et al., 2024c) | 39.0 | **3.44** | 25.7 | 66.0 | **3.04** | 51.3 | 66.0 | 57.3 | 58.6 | 63.3/69.7 | 58.2 | **83.2** | 67.9 |
| **Ours, LLaVAction-7B** | **40.2** | 3.34 | 26.1 | 66.9 | 3.01 | 55.6 | 66.1 | 59.0 | 61.1 | 63.9/71.4 | 58.6 | 82.8 | 70.2 |
| Relative improvement of ours over the baseline LLaVA-Video-7B | +1.2 | -0.1 | +0.4 | +0.9 | -0.03 | +4.3 | +0.1 | +1.7 | +2.5 | +0.6/+1.7 | +0.4 | -0.4 | +2.3 |

Table 7: **Performance on other MLLM benchmarks** that contain human actions. Please note, we are not claiming SOTA, we are noting that we can improve performance over our baseline open-source model (LLaVA-Video-7B (Zhang et al., 2024c)). We also show sub-task performances in Appendix G. We show top-performance closed-source models for reference. Top open-source models are shown in bold, and the second-best are underlined.

as the base MLLMs obtained much better performance (61.0/58.7) compared to the original models (30.5/28.1) and models finetuned with random choices (46.7/43.9) (Table 6). Furthermore, our method beats the baseline (VideoMAE) for both base models.

## 4.4 RESULTS ON OTHER MLLM BENCHMARKS

Apart from comparing with action recognition models, we want LLaVAction to keep general video understanding abilities and also improve fine-grained action understanding on other zero-shot MLLM benchmarks. Therefore, we tested LLaVAction-7B on 13 MLLM benchmarks that test various MLLM video understanding abilities. The evaluated benchmarks consist of two video caption benchmarks, five open-ended Q&A benchmarks and six multi-choice Q&A benchmarks. LLaVAction-7B outperforms LLaVA-Video-7B on 10 benchmarks, indicating the enhanced video understanding ability of our model (Table 7).

## 4.5 ATTENTION-BASED ANALYSIS

We sought to analyze the impact of action-related training and LLaVAction model design in an interpretable manner. Following the approach in (Zhang et al., 2025b), we employed token attention analysis to understand model behavior (Figure 4). We computed average text-visual token correlations for both LLaVA-Video-7B and LLaVAction-7B using the EPIC-KITCHENS-100-MQA dataset. For fair comparison, we fine-tuned LLaVA-Video-7B with randomly generated answer choices. Analysis Methodology: We first computed the text-visual attention tensors of size $N \times T \times V$, where $N$ is the number of data samples, $T$ is the number of text tokens and $V$ is the number of visual tokens. We then calculated the maximum across the text token dimension, followed by computing mean and 90th percentile values to estimate text-visual correlations.

Text-Visual Correlation Results: LLaVA-Video-7B achieved mean and 90th percentile values 0.00476 and 0.0104, respectively, while LLaVAction-7B achieved mean and 90th percentile values 0.00769 and 0.0175, respectively. This indicates that LLaVAction attends more strongly to visual cues compared to LLaVA-Video, likely due to our hard example mining strategy.

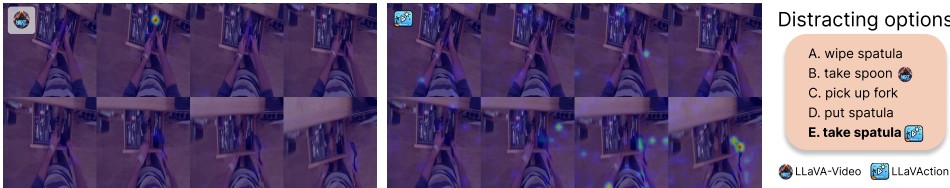

Figure 4: **Qualitative attention for one clip.** Anecdotally, LLaVA-Video mainly attends to the wooden spatula that is placed in the drawer, LLaVAction also attends to the arms and, correctly, the plastic spatula that is being taken. We quantify visual-text token correlations in the main text.

Action Token Analysis: We computed the text-action token attention tensors of size $N \times T \times 1$, calculating the maximum over the text dimension followed by the mean to estimate text-action correlations. LLaVAction demonstrates an average text-action token correlation of 0.143, significantly higher than its text-visual correlation (0.00769). Notably, 99% of visual tokens exhibit lower text correlations compared to action tokens. Since action tokens exclusively attend to visual tokens, this suggests they effectively aggregate visual information relevant to fine-grained actions, enhancing LLaVAction's action understanding capabilities.

### 4.6 DIFFICULTY LEVELS OF FINE-GRAINED ACTION UNDERSTANDING ANALYSIS

While EPIC-KITCHENS-100-MQA features distinguishing fine-grained actions, individual samples may still vary in difficulty. To understand how baseline models and LLaVAction perform across different difficulty levels, we adopted the concept of psychometric curves (Boring, 1917). We utilized GPT-4o to rate the difficulty of distinguishing between the options (detailed prompt in Appendix C.6) from 1 (very easy) to 4 (hard) and then reported the performances of LLaVA-Video and LLaVAction

| Difficulty levels | 1 | 2 | 3 | 4 |
|---|---|---|---|---|
| LLaVA-Video Acc. | 0.405 | 0.360 | 0.342 | 0.339 |
| LLaVAction Acc. | 0.735 | 0.744 | 0.738 | 0.726 |
| # Samples | 400 | 3529 | 2836 | 2903 |

Table 8: **Model performance under different difficulty levels.** LLaVAction is more robust with different semantic similarity between options.

under different levels on EPIC-KITCHENS-100-MQA (Table 8). LLaVAction not only achieves higher overall accuracy but also maintains more robust performance as difficulty increases, showing less performance degradation compared to baseline models.

## 5 CONCLUSION

Recent advances in MLLMs prompted our investigation of their fine-grained action understanding abilities. Through our proposed EPIC-KITCHENS-100-MQA benchmark, which uses similar actions as distractors, we reveal that state-of-the-art MLLMs face significant challenges in action discrimination tasks. We address these limitations by introducing specialized data reformulation strategies and action-aware architectural components that substantially enhance MLLM action recognition capabilities. The resulting LLaVAction model achieves robust performance and demonstrates strong generalization across our benchmark, three additional action recognition datasets, and ten comprehensive MLLM video understanding benchmarks.

## ACKNOWLEDGMENTS

We thank the Swiss AI Initiative Project ID a03 and a144 from the Swiss National Supercomputing Centre (CSCS); H.Q. and A.M. thank the Boehringer Ingelheim Fonds PhD stipend; M.W.M. thanks the Vallee Foundation; M.W.M. and A.M. thank the SNSF by grant No. 320030-227871.

## AUTHOR CONTRIBUTIONS

Conceptualization: S.Y., H.Q., A.M., M.W.M.; Methodology & Software: H.Q., S.Y., A.M., M.W.M.; Experiments: H.Q., S.Y.; Writing: H.Q., S.Y., M.W.M, A.M.; Visualization: H.Q., S.Y., M.W.M.; Funding acquisition: M.W.M., A.M.

ETHICS STATEMENT

This work utilizes established human activity benchmarks and trains the resulting LLaVAction model exclusively on data that contain common daily activities. Given the benign nature of the training data and the focus on routine human behaviors, we anticipate minimal ethical concerns. All human activity data used come from publicly available datasets that have previously been used by the research community.

REPRODUCIBILITY STATEMENT

To ensure reproducibility, we released our code, data, the benchmark, and models at https://github.com/AdaptiveMotorControlLab/LLaVAction.

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

## A  METHODS: LICENSING INFORMATION

| Code/Dataset | License |
|---|---|
| LLaVA-NeXT | Apache-2.0 |
| AVION | MIT |
| TIM | CC-NC-SA-4.0 |
| EPIC-KITCHENS-100 | |
| Dataset 55 and extended | NC-Government |
| EPIC-KITCHENS-100 | |
| Annotations | CC-NC-SA-4.0 |
| lmms-eval | MIT and Apache-2.0 |

Table 9: **List of codes and datasets with their corresponding licenses.**

## B  METHODS: GPT-4O DISTILLATION.

Due to the cost, we sample 4 frames per annotated video clip to go over the training set of EPIC-KITCHENS-100.

We first get the caption corresponding to all video clips in the training set, and then we use the captions obtained to create open-ended question-answers. We show the corresponding prompts for the generations of captions and open-ended question-answers as follows.

### B.1  GPT-4O AND GPT-4O-MINI ANNOTATION PROMPT FOR THE CAPTION TASK.

```
You are viewing video frames from an egocentric perspective and
you are the person.  Describe the video frames in detail and
reason about the actions you are performing.  You will be provided
with the human-annotated ground-truth for the action, but you
should independently come to your own conclusion.

If you disagree with the human annotation, indicate "true" in
the "disagree_with_human_annotation" field of your response,
and provide your reasoning without mentioning the ground-truth
answer.  This will keep your reasoning clean.  If you
agree with the human annotation, indicate "false" in the
"disagree_with_human_annotation" field and provide your reasoning
without referencing the ground-truth to maintain a clean
description.  The true ground-truth action is {gt_answer}.  Your
reasoning steps should include supporting evidence for the action,
such as the duration of the video, the sequence of actions the
person performs, the objects they interact with, and the overall
context of the video.

As a general guideline, for videos longer than 3 seconds,
provide detailed reasoning steps, and for videos shorter than 3
seconds, generate less detailed reasoning.  The video duration
is {end_second - start_second:.3f} seconds.  Make sure you use the
first-person perspective in your reasoning.
```

### B.2  GPT-4O AND GPT-4O-MINI ANNOTATION PROMPT FOR OPEN-ENDED QUESTION ANSWERING

```
Your job is to create 3 question-answer pairs based on the text
below.  The text contains a first-person narrative of video
frames from an egocentric perspective of a person interacting with
objects in a kitchen.  caption_text You can ask questions such
```

```
as: What object am I interacting with? What objects are visible
in the video? What is the sequence of the atomic actions I am
performing? Make sure your questions can be answered based on
the information provided in the text. Do not ask questions that
require additional context or information beyond what is given.
```

# C  METHODS: LLAVACTION TASK PROMPTS

## C.1  LLAVACTION CAPTION PROMPT

```
Describe in details what you see from the video frames. Try to
focus on what you are doing.
```

## C.2  LLAVACTION PERSPECTIVE PROMPT

**Egocentric.** For the EgoSchema benchmark, given that our LLaVAction-7B is trained with egocentric perspective prompt on EPIC-KITCHENS-100, we use the same egocentric perspective prompt when we evaluate our model on EgoSchema benchmark.

```
You are seeing this video from egocentric view and you are the
person. Your hands are sometimes interacting with objects. What
action are you doing?
```

**Allocentric.**

```
The video is taken from egocentric view. The person's hands are
sometimes interacting with objects. What action is the person
doing?
```

## C.3  LLAVACTION TEMPORAL DETECTION PROMPT

```
The provided video contains an action {ACTION NAME} that lasts
2.96 seconds. What is the relative start and end time of the
action in seconds? Format it as 'start_timestamp: end_timestamp'
and round to 2 decimal places.
```

## C.4  LLAVACTION DIRECT PREDICTION PROMPT

```
What action are you performing? Give a short sentence such as
'move knife'.
```

## C.5  LLAVACTION PRIOR ACTION LEARNING PROMPT

```
{prev2_offset} seconds ago, you started an action
{prev2_narration}. {prev1_offset} seconds ago, you started
an action {prev1_narration}. What action are you currently
performing? Here are the options of actions you can select:
```

## C.6  DIFFICULTY LEVEL ASSESSMENT PROMPT

```
You are analyzing a multiple-choice question for fine-grained
action recognition. Your task is to rate the difficulty of
distinguishing between the options based on how similar they are
to each other and the ground truth answer.
```

```
Ground Truth Answer: gt_answer
```

```
Options: options_text
```

```
Please analyze the semantic similarity between the options and
rate the difficulty on a scale of 1-4:
```

```
- 1 (Very Easy):  Options are very different from each other and
the correct answer is obvious

- 2 (Easy):  Options have clear differences, correct answer is
fairly obvious

- 3 (Medium):  Options have moderate similarity, requires some
careful consideration

- 4 (Hard):  Options are quite similar, subtle differences make it
challenging

Consider factors like:

- Semantic similarity between action descriptions

- Specificity vs generality of actions

- Whether options describe similar but distinct actions

- How confusable the distractors are with the ground truth

Respond with only a single number (1-4) representing the
difficulty score.
```

## D  AVION AS OOD DISTRACTORS

We utilized TIM's (Chalk et al., 2024) predictions to build our EPIC-KITCHENS-100-MQA benchmark. When we use similar ideas to build the hard distractor training set, it results in IID setting if we still use TIM's (Chalk et al., 2024) predictions. Methods could directly overfit on TIM's choice combination to obtain better performances instead of contrasting fine-grained actions. Therefore, we used AVION's (Zhao & Krähenbühl, 2023) predictions during training to serve as an OOD setting.

Here, we provide three experiments to support using AVION can be an OOD setting. First, we calculated the top-1 agreement percentage (65%) and top-5 overlap percentage (45%) between AVION and TIM in EPIC-KITCHENS-100's validation set, suggesting a considerable difference in distribution, especially when K is larger.

Additionally, we computed the Jensen–Shannon Divergence (JSD) between the softmax outputs of Avion and Tim across the validation set (9668 samples). The mean JSD was $0.674 \pm 0.089$, with a 95% confidence interval of $[0.672, 0.765]$. We obtained a p-value $< 0.001$ and a Cohen's $d$ of 7.57, indicating a large and statistically significant difference between Avion and Tim. As a consequence, their generated distractors should be seen as coming from two different distributions.

An important signature of the IID vs. OOD argument is that OOD is less vulnerable to overfitting when giving the model more chances to explore the training data. Following our OOD setting that uses Avion distractors for training and TIM distractors for testing, we performed experiments that vary K in both training and testing, for K = 5, 10, 20, we got 74.3, 69.5, 64.1, respectively. Since test-time distractors are generated by TIM and training-time distractors are generated by AVION, we believe increasing K in training introduces overfitting, and thus it does not generalize well to TIM's distractors, which further supports our rationale for the OOD setting.

## E  ADDITIONAL ABLATIONS

### E.1  ACTION TOKEN DESIGN ABLATION

Our action token design (Section 3.3) effectively sees and encodes action-related video information, and hence benefits the question-answering task. To further support our action token design, we implement several other token-aggregation methods. Specifically, our action token is one learnable token added between the visual tokens and text tokens and is supervised in the last layer with the action classification loss. We hence implement three variants: 1) adding three action tokens and supervising them with verb, noun, and action separately in the last layer; 2) adding one action token and supervising it in the first layer of the MLLM; adding one action token and supervising it across

all MLLM layers. The results are in Table 10 on our benchmark. We can see our action design performs the best while keeping simplicity.

| Token designs | Acc. |
|---|---|
| 3 tokens, last layer | 68.8 |
| 1 token, first layer | 66.2 |
| 1 token, all layers | 31.7 |
| 1 token, last layer (Ours) | **69.1** |

Table 10: **Action token design ablation study**. The top 1 percent accuracy is shown.

### E.2 LEAVE-ONE-OUT ABLATION

Remarkably, the 10-point gain over our baseline model cannot be attributed to only a single factor. We took our full model, i.e., the base plus all added methods, which we call **LLaVAction-7B**, and performed a leave-on-out ablation (Table 11). Given our additions adds negligible overhead in the inference time (only one special vision token added to the baseline model), we then suggest using our full LLaVAction-7B and techniques in downstream tasks.

| LLaVA-Video-7B | Acc. |
|---|---|
| Full (LLaVAction-7B) | 74.1 |
| Full w/o adversarial distractors (AVION) | 69.7 |
| Full w/o action token | 73.6 |
| Full w/o temporal detection | 72.2 |
| Full w/o GPT-4o-based reformulation | 73.2 |
| Full w/o direct prediction | 73.2 |
| Full w/o temporal order learning | 72.3 |

Table 11: **Leave-one-out ablation study**. Full denotes having all the proposed methods. In each row we drop one method from the full method and report the resulting performance. 16 frames were used for both training and testing, and the percent accuracy is shown.

### E.3 STRUCTURED ACTION PREDICTION ABLATION

Our proposed two-stage pipeline enables MLLMs to fairly compare and outperform other SOTA action recognition methods. MLLMs directly output free texts, which makes it hard to find an exact match with the action labels in the dataset, especially when the action space is huge and fine-grained. With an external model applied in the first stage to filter out easy, irrelevant actions, MLLMs can mainly focus on differentiating between the hard distracting actions. To support that, we evaluated LLaVA-Video-7B with the same external model on the EPIC-KITCHEN action recognition benchmark (Table 12). We can see that the performance of LLaVA-Video-7B is much worse, even with an external model, showing that it struggles to solve the hard distractors. Meanwhile, we further implement another way (denoted as 'Multi-round appending') to achieve structured action output. Specifically, we first prefill and store the KV Cache for video+prompt+question to avoid repeated computation. After that, we append each action class to compute the text cross-entropy loss. The action class with the lowest loss is selected as the final action prediction. We test for both the zero-shot LLaVA-Video-7B and our LLaVAction models on the EPIC-KITCHEN action recognition benchmark (Table 12). The results show our two-stage action prediction pipeline can obtain much better performance under both fine-tuned and zero-shot settings. Most importantly, the multi-round appending is extremely time-consuming. Although KV Cache storage avoids computing video+prompt+question repeatedly, the model still needs to infer 4K times to obtain the correct answer. Evaluating the model on EPIC-KITCHENS-100's validation set (9668 samples) takes around 820 GPU hours when using the multi-round appending approach. In comparison, our two-stage approach only takes 4.3 GPU hours, making our two-stage method 190 times faster.

| Methods | Acc. |
|---|---|
| *LLaVA-Video-7B* | |
| Multi-round appending | 10.3 |
| Two-stage (Ours) | 26.5 |
| *LLaVAction-7B* | |
| Multi-round appending | 40.0 |
| Two-stage (Ours) | **58.3** |

Table 12: **Structured action prediction ablation study**. The top 1 percent accuracy is shown.

| Methods | Action label | Action narration |
|---|---|---|
| zero-shot LLaVA-Video-7B | 26.5 | 35.7 |
| zero-shot LLaVA-OV-7B | 19.6 | 28.9 |
| zero-shot LLaVA-OV-0.5B | 24.8 | 32.0 |

Table 13: **Quantitative results for action label vs. action narration.** Models are inferred with eight frames as inputs.

## F   QUALITATIVE EXAMPLES

### F.1   COMPARING ACTION NARRATION AND ACTION LABEL IN EPIC-KITCHENS-100

The action labels in EPIC-KITCHENS-100 originate from the raw action narrations that are curated and compressed by a combination of word clustering and iterative manual refinement (Damen et al., 2022). However, this compression might change the semantic meaning of both nouns, verbs and the way they are combined. As a result, large language models that are sensitive to the meaning of words can be misled (see comparisons in Appendix Figure 5 and Table 13). While we show SOTA results using the action label, we note that we can achieve better performance if we use the uncompressed, original narrations. We hope that our work could inspire future work to study the best text representation of actions to train and evaluate MLLMs in action recognition.

Qualitatively: we illustrate some examples of choices represented in the action label manner (Appendix Figure 5). We show the ground truth option in blue and the prediction of LLaVAction-7B in pink. We can see LLaVAction-7B's predictions also make sense in those examples and hence cause ambiguity across choices. Instead, the corresponding action narration fits better to the language's nature and can better describe the video content with less ambiguity.

Quantitatively: furthermore, we also quantify MLLMs' zero-shot performance (LLaVA-OV-0.5B, LLaVA-OV-7B, LLaVA-Video-7B) when using action labels or action narrations as inputs (Table 13). The inferior zero-shot performance of all 3 evaluated models when tested on the action labels as action representation supports our qualitative observations that action labels are less ideal than narrations for MLLMs.

### F.2   DIFFERENT CHOICES COMPARISON

Here, we show examples of choices generated by random sampling, AVION top-5 predictions, and TIM top-5 predictions (Appendix Figure 6). We can see that the randomly selected choices have many trivial choices that can be easily distinguished with the correct answer. In comparison, choices generated based on AVION and TIM top-5 predictions become much more similar to the correct answer and exhibit features such as similar object/scene, temporal orders or object relationships that are emphasized by other benchmarks.

### F.3   LLAVACTION CAPTION

Here, we show one video caption example of different models including GPT-4o, LLaVA-Video-7B and our LLaVAction-7B (Appendix Figure 7). We can see the interacting object (pizza piece) is

pretty small in the video and there are also many other distracting objects. Both GPT-4o and LLaVA-Video-7B cause 'hallucinations' in their descriptions. For example, GPT-4o thinks the person holds the slice with both hands. Instead, LLaVAction-7B still retains the video caption ability and can generate plausible descriptions of the video.

## G    SUB-CATEGORY PERFORMANCE COMPARISONS ON THE ADDITIONAL BENCHMARKS

Snice MVBench and LongVideoBench also have sub-category measurements with many of them related to action understanding, we also show the sub-category performances on these two benchmarks in this section.

### G.1    PERFORMANCE COMPARISON ON SUB-CATEGORIES OF MVBENCH

Here we show the performance comparison between LLaVA-Video-7B and our LLaVAction-7B on sub-categories of MVBench. We can see LLaVAction-7B boost the performance on many action-related categories such as action count, action sequence and fine-grained action, etc.

| Tasks | LLaVA-Video-7B | LLaVAction-7B (Ours) | Difference |
|---|---|---|---|
| Action antonym | 76.0 | 75.0 | -1.0 |
| Action count | 57.0 | 65.0 | 8.0 |
| Action localization | 61.0 | 63.5 | 2.5 |
| Action prediction | 62.0 | 59.0 | -3.0 |
| Action sequence | 70.5 | 72.5 | 2.0 |
| Character order | 74.5 | 79.0 | 4.5 |
| Counterfactual inference | 50.0 | 52.0 | 2.0 |
| Egocentric navigation | 30.5 | 28.0 | -2.5 |
| Episodic reasoning | 53.5 | 54.0 | 0.5 |
| Fine-grained action | 48.0 | 49.0 | 1.0 |
| Fine-grained pose | 54.5 | 61.5 | 7.0 |
| Moving attribute | 71.0 | 72.5 | 1.5 |
| Moving count | 44.0 | 43.0 | -1.0 |
| Moving direction | 35.5 | 31.0 | -4.5 |
| Object existence | 60.0 | 59.0 | -1.0 |
| Object interaction | 84.5 | 83.5 | -1.0 |
| Object shuffle | 41.5 | 44.0 | 2.5 |
| Scene transition | 93.5 | 90.5 | -3.0 |
| State change | 54.0 | 61.5 | 7.5 |
| Unexpected action | 81.5 | 79.0 | -2.5 |

Table 14: **Sub-category comparison with LLaVA-Video-7B on MVBench.**

### G.2    PERFORMANCE COMPARISON ON SUB-CATEGORIES OF LONGVIDEOBENCH

Here we show the performance comparison between LLaVA-Video-7B and our LLaVAction-7B on sub-categories of LongVideoBench. We can see LLaVAction-7B also boosts the performance on many action-related categories such as event before/after, text-referred object attribute, and object-before/after object.

### G.3    PERFORMANCE COMPARISON ON SUB-CATEGORIES OF VIDEOMME

Here we show the performance comparison between LLaVA-Video-7B and our LLaVAction-7B on sub-categories of VideoMME. Our model did not improve the action recognition performance on VideoMME possibly due to the domain gap between VideoMME and EPIC-KITCHENS-100.

| Tasks | LLaVA-Video-7B | LLaVAction-7B (Ours) | Difference |
|---|---|---|---|
| Event-Referred object | 72.31 | 69.23 | -3.08 |
| Event-Before/after event | 67.02 | 67.02 | 0.0 |
| Object-Referred event | 67.82 | 64.37 | -3.45 |
| Object-Before/after object | 57.58 | 59.09 | 1.52 |
| Scene-Referred object attribute | 71.59 | 70.46 | -1.14 |
| Scene-Referred event | 72.04 | 66.67 | -5.38 |
| Scene-Referred object | 63.89 | 63.89 | 0.0 |
| Scene-Referred object attribute change | 55.56 | 52.78 | -2.78 |
| Scene-Referred object tracking | 65.43 | 66.67 | 1.23 |
| Sequence of scenes | 41.24 | 41.24 | 0.0 |
| Text-Referred object attribute | 59.49 | 62.02 | 2.53 |
| Text-Referred event | 56.92 | 56.92 | 0.0 |
| Text-Referred object | 59.21 | 59.21 | 0.0 |
| Event before/after text | 50.68 | 58.90 | 8.22 |
| Object before/after text | 58.11 | 52.70 | -5.41 |
| Text-Referred object attribute change | 47.56 | 50.00 | 2.44 |
| Text-Referred object tracking | 32.88 | 32.88 | 0.0 |

Table 15: **Sub-category comparison with LLaVA-Video-7B on LongVideoBench.**

# H    EXTENDED DISCUSSION

**Egocentric vs. Allocentric perspective.** MLLM can be assigned with different roles before seeing the video. Since the videos are taken from the first-person perspective, we believed the egocentric perspective aligns better with the LLM pretraining data. Therefore, we switch from the third-person (allocentric) perspective to the first-person (egocentric) perspective to better guide the model. We present the prompts in the Appendix C.2. When we fix the distractors from random sampling, using the egocentric prompt gives a 0.5 point improvement over using the allocentric prompt on EPIC-KITCHENS-100-MQA.

**Alternative approaches we tested.** We also tested a few alternative approaches to improve MLLMs in our benchmark. We tried self-consistency predictions, which do not yield improvements, perhaps due to the task being vision-centric. Additionally, we explored multi-modal chain-of-thought (COT) reasoning by prompting the model to generate a caption prior to addressing the multi-question answering task. However, we found that the model exhibited reluctance to perform this action, despite being capable of generating captions or answering multi-choice questions independently. A variant of it is to inference the model twice, so we have the caption first and feed that into the instruction of answering multi-choice question task, similar to (Zhang et al., 2024d). While a minor improvement was observed, we think it is not worth the 2X compute. We believe that video action recognition is a good way to explore video reasoning for MLLMs. However, we leave COT improvements on this task for future work.

| Tasks | LLaVA-Video-7B | LLaVAction-7B (Ours) | Difference |
|---|---|---|---|
| Categories: Artistic Performance | 68.9 | 69.4 | 0.5 |
| Categories: Film & Television | 71.7 | 72.8 | 1.1 |
| Categories: Knowledge | 76.0 | 75.4 | -0.6 |
| Categories: Life Record | 71.7 | 71.7 | 0.0 |
| Categories: Multilingual | 67.8 | 61.1 | -6.7 |
| Categories: Sports Competition | 65.6 | 66.4 | 0.8 |
| Task Categories: Action Reasoning | 69.5 | 69.1 | -0.4 |
| Task Categories: Action Recognition | 69.0 | 68.4 | -0.6 |
| Task Categories: Attribute Perception | 83.8 | 83.3 | -0.5 |
| Task Categories: Counting Problem | 48.1 | 46.3 | -1.8 |
| Task Categories: Information Synopsis | 87.0 | 86.7 | -0.3 |
| Task Categories: OCR Problems | 73.4 | 74.1 | 0.7 |
| Task Categories: Object Reasoning | 73.6 | 73.3 | -0.3 |
| Task Categories: Object Recognition | 75.7 | 77.1 | 1.4 |
| Task Categories: Spatial Perception | 68.5 | 72.2 | 3.7 |
| Task Categories: Spatial Reasoning | 82.1 | 82.1 | 0.0 |
| Task Categories: Temporal Perception | 76.4 | 78.2 | 1.8 |
| Task Categories: Temporal Reasoning | 51.4 | 52.0 | 0.6 |
| Video Sub Categories: Acrobatics | 65.6 | 64.4 | -1.2 |
| Video Sub Categories: Animation | 58.9 | 60.0 | 1.1 |
| Video Sub Categories: Astronomy | 77.8 | 77.8 | 0.0 |
| Video Sub Categories: Athletics | 66.7 | 73.3 | 6.6 |
| Video Sub Categories: Basketball | 54.4 | 51.1 | -3.3 |
| Video Sub Categories: Biology & Medicine | 78.9 | 78.9 | 0.0 |
| Video Sub Categories: Daily Life | 78.9 | 75.6 | -3.3 |
| Video Sub Categories: Documentary | 74.4 | 76.7 | 2.3 |
| Video Sub Categories: Esports | 62.2 | 60.0 | -2.2 |
| Video Sub Categories: Exercise | 58.9 | 67.8 | 8.9 |
| Video Sub Categories: Fashion | 68.9 | 70.0 | 1.1 |
| Video Sub Categories: Finance & Commerce | 80.0 | 80.0 | 0.0 |
| Video Sub Categories: Football | 72.2 | 75.6 | 3.4 |
| Video Sub Categories: Geography | 76.7 | 75.6 | -1.1 |
| Video Sub Categories: Handicraft | 77.8 | 76.7 | -1.1 |
| Video Sub Categories: Humanity & History | 66.7 | 67.8 | 1.1 |
| Video Sub Categories: Law | 82.2 | 80.0 | -2.2 |
| Video Sub Categories: Life Tip | 70.0 | 67.8 | -2.2 |
| Video Sub Categories: Literature & Art | 80.0 | 73.3 | -6.7 |
| Video Sub Categories: Magic Show | 62.2 | 65.6 | 3.4 |
| Video Sub Categories: Movie & TV Show | 68.9 | 70.0 | 1.1 |
| Video Sub Categories: Multilingual | 67.8 | 61.1 | -6.7 |
| Video Sub Categories: News Report | 84.4 | 84.4 | 0.0 |
| Video Sub Categories: Other Sports | 72.2 | 72.2 | 0.0 |
| Video Sub Categories: Pet & Animal | 78.9 | 78.9 | 0.0 |
| Video Sub Categories: Stage Play | 82.2 | 80.0 | -2.2 |
| Video Sub Categories: Technology | 72.2 | 77.8 | 5.6 |
| Video Sub Categories: Travel | 75.6 | 77.8 | 2.2 |
| Video Sub Categories: Variety Show | 65.6 | 67.8 | 2.2 |

Table 16: **Sub-category comparison with LLaVA-Video-7B on VideoMME.**

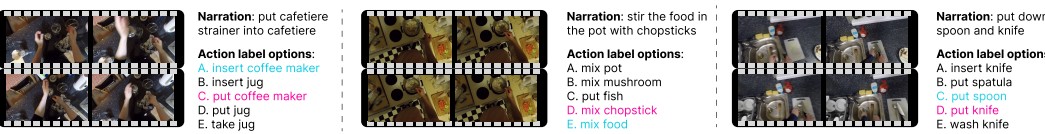

Figure 5: **Action labels vs. narrations.** Blue option denotes ground truth and the pink option denotes LLaVAction-7B's prediction. Action labels usually reduce multiple nouns into one noun, resulting in ambiguity that could mislead a MLLM. Note that the narration also contains crucial particles with the phrasal verbs to clarify the meaning such as "put down", "put into".

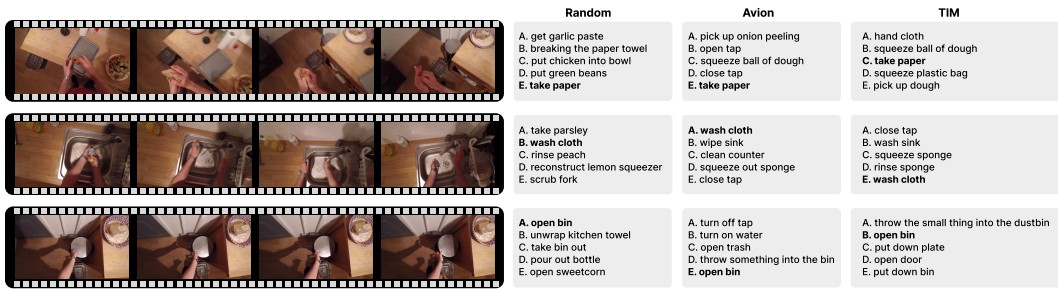

Figure 6: **TIM's choices are harder than AVION and random** by introducing more visually similar objects and actions. Bold option denotes ground truth.

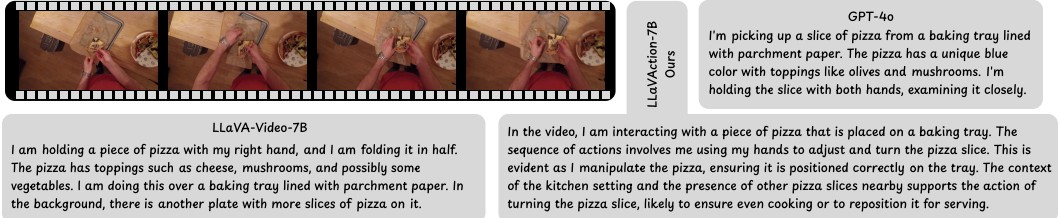

Figure 7: **The captioning capability can provide insights into models.** A comparison among LLaVAction-7B, GPT-4o and LLaVA-Video-7B.

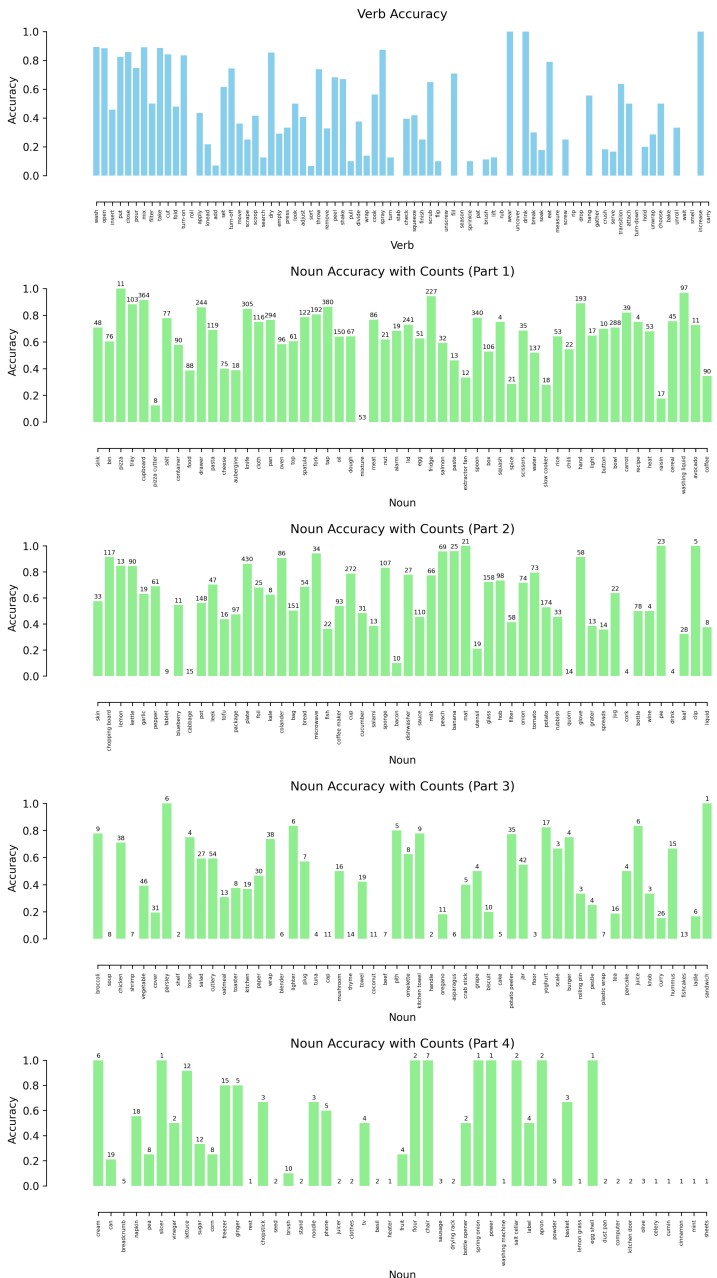

Figure 8: **Breakdown of the performance of LLaVAction-7B on Verbs and Nouns.** We analyzed the accuracy per verb and noun in EPIC-KITCHENS-100 for that our LLaVAction-7B (32f), evaluated on the validation set with action labels (i.e., the model reported in Table 4 that achieved 58.3 accuracy). There are more nouns than verbs, thus nouns are shown across four subplots for visualization but otherwise are not separated in an intentional way. The number above each bar is the total per class.

