# OpenReview forum: "LLaVAction: evaluating and training multi-modal large language models for action understanding"
_ICLR.cc/2026/Conference — ICLR 2026 Poster_

### Official Review · Reviewer_BpZo · 2025-10-31

**Soundness:** 2
**Presentation:** 3
**Contribution:** 2
**Rating:** 4
**Confidence:** 4

**Summary:**

The technical contributions, especially the comprehensive training data mixture, clearly demonstrate improved performance over baselines like GPT-4o.

However, the effectiveness of the new benchmark is weakened by concerns regarding the **validity of its hard negative samples**, which undermines the strength of the evaluation methodology.
Furthermore, the specialized Action Token unnecessarily introduce task-specific constraints that **limit the model’s generalization capability**. The two-stage structured action prediction pipeline is an **unnecessary design**.

Given that the benchmark validity is questionable and the proposed methods either lack necessity or generality, but recognizing the significant technical gains achieved, the paper lands at a borderline reject.

**Strengths:**

LLaVAction integrates structured training tasks like temporal detection and temporal order learning to teach the model action boundaries and natural continuity. Such training data mixture strategy improves action understanding ability of MLLMs.

**Weaknesses:**

The Hard Example Mining strategy used for the new EPIC-KITCHENS-100-MQA benchmark is questionable due to the validity of hard negatives.
The generated "hard distractors"—which are chosen for being highly confusable errors—may be semantically correct actions due to potential annotation ambiguity/near synonyms, or missing labels. The EPIC-KITCHENS dataset has 4K action labels, which is hard to annotate exhaustively in the original dataset.
Although the mechanism guarantees the exclusion of the exact ground truth class ID, it does not ensure the distractors are true semantic negatives.



The model’s Action Token design compromises MLLM generality by employing explicit classification heads to predict a finite set of nouns and verbs. While intended to enhance visual feature utilization, this hard-coded classification is specialized for the datasets with closed set labels, moving away from the flexible, open-vocabulary text generation paradigm characteristic of general MLLMs.



The proposed two-stage structured action prediction pipeline is unnecessary.
The authors contend that using this pipeline is essential because "One could put all the possible actions in the question prompt and let MLLMs choose, but this will become impossible when the number of action types increases" (e.g., 4k actions). However, this argument is questionable because simpler, direct classification methods exist that avoid context overflow, such as storing the video+prompt+question in the KV Cache and calculating the loss by appending individual class names to perform 4K classification natively.

**Questions:**

The authors could discuss more about the Hard Example Mining strategy, since the other shortcomings are hopeless.

---

> ### Author Response · Authors · 2025-11-24
> **Response to Reviewer BpZo (1)**
>
> We thank the reviewer for their thoughtful comments and the time and effort spent reviewing our paper. We appreciate that they recognized the effectiveness of our action-related MLLM data reformulation pipeline. Below are our responses regarding the concerns.
>
> > 1. “The Hard Example Mining strategy used for the new EPIC-KITCHENS-100-MQA benchmark is questionable due to the validity of hard negatives.”, “The authors could discuss more about the Hard Example Mining strategy”.
>
> - First, the hard negatives are by definition from valid, supposedly independent classes of action recognition models (Eq. 3) and EPIC-KITCHENS-100. These hard negative classes are part of what makes the original EPIC-KITCHENS-100 challenging and interesting. EPIC-KITCHENS-100’s action annotations are carefully designed and verified by humans, which makes it one of the most existing valid datasets to improve and evaluate models’ fine-grained action understanding ability. A lot of efforts from the community have been made to push models to be better at distinguishing correct actions from hard wrong ones, and our effort is part of it by using a two-stage pipeline and hard example mining with MLLMs and SOTA specialized models.
>
> - Secondly, annotation errors or miss-labels could happen in EPIC-KITCHENS-100’s annotation, but that generally happens for all the datasets and benchmarks. However, a small portion of errors does not influence the main goal of the benchmark, i.e., to compare different models’ fine-grained action understanding ability, since all the models, especially zero-shot models, will answer wrongly for those erroneous samples. In that sense, GPT-4o, GPT-4o-mini and LLaVA-Video-7B obtain 52.2, 37.4 and 35.9 top 1 accuracy separately on our benchmark (Figure 1), all of which outperform chance level (20%) and their performances in our benchmark positively correlate with public MLLM benchmarks. This supports that the hard negative samples are not just confusable errors and could correctly reflect models’ performance. Apart from zero-shot evaluation, the large improvements on EPIC-KITCHENS-100-MQA and positive transfers across a broad range of video benchmarks after training with our data reformulation pipeline further support the effectiveness of the hard negative samples.
>
> - Thirdly, instead of using the raw action labels, we construct the EPIC-KITCHENS-100-MQA benchmark using action narrations, which can give more accurate action descriptions to avoid confusion (please see Figure 3 for our benchmark examples and Figure 5 for comparison between action labels and action narrations).
>
> - Lastly, we respect the reviewer’s concern about EPIC-KITCHENS-100’s annotation quality. However, our core contribution, the hard example mining strategy to evaluate and improve MLLM’s fine-grained action understanding ability, could be applied to any other domain or dataset that has better annotation quality. In the manuscript, we have already extended to the tool-assembling Meccano dataset and the very recent EPFL-SmartKitchen-30 dataset. To further support the effectiveness of our hard example mining strategy, we tested our method on a very different domain, animal fine-grained behavior understanding. Specifically, we tested on the Animal Kingdom dataset [1], which has 140 fine-grained actions and at most 12 actions can happen at the same time (multi-classification task). We adapted LLaVA-Video-7B and used VideoMAE to generate hard examples. The results are as follows, reported in Jaccard accuracy. We can see LLaVAction can obtain much better performance compared to zero-shot and finetuned with random choice. And LLaVAction can further beat the baseline VideoMAE.
>
> | Methods                | Jaccard Acc.  |
> |-----------------------|------------------|
> | VideoMAE           | 53.13              |
> | LLaVA-Video-7B (zero-shot)             | 30.49             |
> | LLaVA-Video-7B (random choice)    | 46.73             |
> | LLaVAction (ours)         | 61.00             |
>
> > 2. “The model’s Action Token design compromises MLLM generality by employing explicit classification heads to predict a finite set of nouns and verbs”
>
> - Our action token only serves as an additional learning objective and is not used to give the final prediction. During inference, or when training with tasks that have no clear action labels (such as video caption, open-ended question answering and general multi-choice question answering), we can simply just compute the final text generation loss. Our evaluation on 13 zero-shot MLLM benchmarks that cover very different tasks and domains shows that the addition of the action token does not weaken the flexible, open-vocabulary text generation paradigm characteristic of general MLLM.

---

> > ### Author Response · Authors · 2025-11-24
> > **Response to Reviewer BpZo (2)**
> >
> > > 3. “The proposed two-stage structured action prediction pipeline is unnecessary… storing the video+prompt+question in the KV Cache and calculating the loss by appending individual class names to perform 4K classification natively. ”
> >
> > - We thank the reviewer for suggesting a variant to compare with our two-stage action prediction pipeline. We implement the suggested method (denoted as ‘*Multi-round appending*’) by first prefilling and storing the KV Cache for video+prompt+question and then appending each action class to compute the text cross-entropy loss. We test for both the zero-shot LLaVA-Video-7B and our LLaVAction models on the EPIC-KITCHEN action recognition benchmark. The results are as follows.
> >
> > | Methods                          | Top1 Acc.  |
> > |-----------------------|------------------|
> > |**LLaVA-Video-7B**|
> > | Multi-round appending   | 10.34         |
> > | Two-stage (Ours)            | 26.49         |
> > |**LLaVAction-7B**|
> > | Multi-round appending  | 40.03         |
> > | Two-stage (Ours)           | 58.3           |
> >
> > - The results show our two-stage action prediction pipeline can obtain much better performance under both fine-tuned and zero-shot settings.
> >
> > - Most importantly, the suggested multi-round appending is extremely time-consuming. Although KV Cache storage avoids computing video+prompt+question repeatedly,  the model still needs to infer 4K times to obtain the correct answer. Evaluating the model on EPIC-KITCHENS-100’s validation set (9668 samples) takes around 820 GPU hours when using the multi-round appending approach. In comparison, our two-stage approach only takes 4.3 GPU hours, making our two-stage method 190 times faster.
> >
> > [1]. Ng, Xun Long, et al. "Animal kingdom: A large and diverse dataset for animal behavior understanding." Proceedings of the IEEE/CVF conference on computer vision and pattern recognition. 2022.

---

> > > ### Author Response · Authors · 2025-11-26
> > > **Kind Reminder: Seeking Reviewer Feedback for Author/Reviewer Discussion Phase**
> > >
> > > Dear reviewer BpZo,
> > >
> > > We hope this message finds you well. We would like to express our gratitude for the time and effort that you have dedicated to the review process of our submission.
> > >
> > > Furthermore, we are writing to kindly remind you that the rebuttal period is coming to a close. Your feedback and evaluation are crucial to the progress of our work, and we value your expertise and insights immensely.
> > >
> > > During the rebuttal, we hope we have addressed all the queries and concerns you raised regarding our submission. Each point has been carefully considered, and we have provided detailed responses.
> > >
> > > We are eager to engage in further discussions with you on the responses and any other aspects of our study. Your feedback plays a pivotal role in shaping the direction of our research, and we are keen to hear your thoughts and suggestions.
> > >
> > > Thank you for your consideration.
> > >
> > > Best regards,
> > >
> > > Authors of submission #9309

---

### Official Review · Reviewer_V9cL · 2025-10-31

**Soundness:** 2
**Presentation:** 3
**Contribution:** 3
**Rating:** 6
**Confidence:** 5

**Summary:**

This study focuses on improving the understanding of complex human actions by emerging multimodal large language models (MLLMs). The researchers reformulate the EPIC-KITCHENS-100 dataset into a new benchmark, EPIC-KITCHENS-100-MQA, to evaluate MLLMs' fine-grained action recognition. They find that leading MLLMs struggle with identifying correct actions, especially when presented with challenging distractors. To address this, they develop a supervised fine-tuning dataset encompassing tasks like hard action recognition, temporal detection, and free-form question answering. They also propose a new model, LLaVAction, which improves attention to visual tokens and utilizes a two-stage pipeline to better understand structured actions. The model significantly enhances performance, achieving notable accuracy gains on both MLLM and action recognition benchmarks, highlighting its potential for tackling complex human behavior tasks.

**Strengths:**

1. The paper is well-written, presenting the methodology and experimental results in a clear and systematic manner.
2. The research topic is highly practical and meaningful. Enhancing MLLMs' ability to better understand human actions could benefit a wide range of real-world applications.
3. By approaching the problem from an action recognition perspective, the paper introduces a new, challenging benchmark. This dataset holds potential to contribute valuable insights to research within the relevant domain.
4. The proposed LLaVAction model not only demonstrates strong performance in action recognition tasks but also achieves improvements across more generic benchmark datasets. This is a noteworthy phenomenon, showcasing the potential value and versatility of the approach to broader applications.

**Weaknesses:**

1. While the paper focuses on action understanding, its evaluation is limited to action recognition tasks. Comprehensive human action understanding involves other dimensions, such as temporal dynamics, the meaning or intent behind actions, and human interactions. Previous works like MotionLLM [1] and HAIC [2], which explore the use of MLLMs for human action understanding beyond just action recognition, have introduced broader benchmarks and methodologies covering these aspects. This type of works are highly relevant to this paper's domain and should be discussed as part of the related work. Furthermore, demonstrating LLaVAction's effectiveness on these more comprehensive action understanding benchmarks could strengthen the proposed method.
2. Although the paper shows that LLaVAction improves performance on general video understanding benchmarks, a detailed analysis of the reasons behind these improvements is missing. Investigating which aspects of the proposed training methodology contribute to these gains would provide deeper insights into the strengths of the approach.

[1] Chen, Ling-Hao, Shunlin Lu, Ailing Zeng, Hao Zhang, Benyou Wang, Ruimao Zhang, and Lei Zhang. "Motionllm: Understanding human behaviors from human motions and videos." arXiv preprint arXiv:2405.20340 (2024).

[2] Wang, Xiao, Jingyun Hua, Weihong Lin, Yuanxing Zhang, Fuzheng Zhang, Jianlong Wu, Di Zhang, and Liqiang Nie. "HAIC: Improving Human Action Understanding and Generation with Better Captions for Multi-modal Large Language Models." ACL 2025.

**Questions:**

Will the datasets, models, and related resources be made publicly available?

---

> ### Author Response · Authors · 2025-11-24
> **Response to Reviewer V9cL**
>
> We thank the reviewer for their thoughtful comments and the time spent on our paper and insightful feedback that helps us make the paper better. We appreciate the reviewer for recognizing the practicality of the work, the contribution of the dataset and the generalization ability of the proposed method. We would like to address the reviewer’s concerns in the following.
>
> > 1. “While the paper focuses on action understanding, its evaluation is limited to action recognition tasks. Comprehensive human action understanding involves other dimensions, such as temporal dynamics, the meaning or intent behind actions, and human interactions.
>
> - While action recognition is one of our main focuses, as it is the fundamental task for MLLM’s fine-grained action understanding ability, we train and evaluate LLaVAction also for other action understanding tasks. For the training stage, our action-related MLLM data reformulation pipeline involves other dimensions, such as temporal dynamics by including temporal detection and context-aware action recognition in our training. We also included GPT-4o annotations that tackled detailed video captions that capture detailed descriptions of the actions, movements and environments from the egocentric view. For the evaluation stage, we tested on 13 zero-shot MLLM benchmarks that include very different action understanding abilities and LLaVAction can obtain consistent improvements.
>
> > 2. “Previous works like MotionLLM [1] and HAIC [2], which explore the use of MLLMs for human action understanding beyond just action recognition, have introduced broader benchmarks and methodologies covering these aspects. This type of works are highly relevant to this paper's domain and should be discussed as part of the related work. Furthermore, demonstrating LLaVAction's effectiveness on these more comprehensive action understanding benchmarks could strengthen the proposed method.”
>
> - We appreciate the reviewer for pointing out those related works, which are highly relevant and we would like to extend the discussion to include these works.
>
> - MotionLLM represents a promising direction to directly include motion inputs by incorporating a motion encoder. It has shown improvements on their proposed MoVidBench as well as additional video benchmarks. However, MotionLLM requires a big change in MLLM’s architecture to use a motion encoder to take motion inputs, and our method is an attempt to keep the original MLLM’s architecture intact except for a minimal addition of the action token while potentially learning temporal dynamics as well.
>
> - HAIC leverages a novel strategy to curate large video-text pairs that capture detailed actions and interactions. Trained on their training dataset, a considerable improvement is made in their proposed HAICBench and additional video benchmarks, highlighting the importance of high quality video-caption dataset for human action understanding.
>
> - Similarly, LLaVAction, MotionLLM and HAIC all propose the training data pipeline and the evaluation benchmarks. However, the main focus of the training data and benchmarks varies. LLaVAction focuses more on fine-grained action contrastiveness. MotionLLM focuses more on human motion understanding. HAIC focuses more on detailed action captions. Comparing the three models’ general action understanding ability on the newly proposed benchmarks is unfair due to the different training sources.    Instead, we should compare other zero-shot benchmarks. MotionLLM and HAIC have shown results on ActivityNetQA, PerceptionTest, and MVBench. LLaVAction can obtain superior or comparable performances on those benchmarks. Moreover, LLaVAction also evaluates on 10 more zero-shot MLLM benchmarks, showing the consistent improvements of LLaVAction.
>
> > 3. “Although the paper shows that LLaVAction improves performance on general video understanding benchmarks, a detailed analysis of the reasons behind these improvements is missing. Investigating which aspects of the proposed training methodology contribute to these gains would provide deeper insights into the strengths of the approach.”
>
> - We provide a detailed sub-category analysis of the video benchmarks in Tables 12 and 13 of the Appendix. Using VideoMME (Table 13) as a representative example, we observe substantial improvements in object recognition, spatial reasoning, temporal perception, and temporal reasoning. This aligns with our expectations, as recognizing fine-grained actions inherently requires stronger capabilities in these sub-tasks. The only notable degradation appears in the multilingual sub-task, which is also not surprising given that our training data contains only English annotations.
>
> > 4. “Will the datasets, models, and related resources be made publicly available?”
>
> Yes, we would release the codebase, all the model weights, the whole benchmark and the re-formulated training data upon acceptance.

---

> > ### Comment · Reviewer_V9cL · 2025-11-26
> > **Response to Rebuttal**
> >
> > Thank you for your detailed response and clarifications. You have addressed all of my concerns. Therefore, I keep my positive rating.

---

> > > ### Author Response · Authors · 2025-11-26
> > >
> > > Dear reviewer V9cL,
> > >
> > > We're encouraged that you found all concerns addressed and really appreciate the positive rating. Your feedback has been invaluable in strengthening our work.
> > >
> > > Thank you!
> > >
> > > Authors of submission #9309

---

### Official Review · Reviewer_quqa · 2025-10-31

**Soundness:** 3
**Presentation:** 4
**Contribution:** 3
**Rating:** 4
**Confidence:** 4

**Summary:**

This paper tackles the challenge of action understanding in Mutli-modal Large Language Models (MLLMs). The paper address this challenge by introducing a new and challenging benchmark EPIC-KITCHENS-100-MQA, which a multiple-choice, fine-grained action-understanding benchmark by reformulating the EK100 datasets into a multiple-choice question-answering (MQA) task. More importantly, the paper makes its contribution by utilizing hard distractors, leveraging predictions from latest action recognition models to generate semantically similar and difficult distraction choices. The paper then proposes LLaVAction, which introduces an “action token” to enhance focus on visual information and a two-stage inference pipeline to produce structured action outputs. On EPIC-KITCHENS-100-MQA, LLaVAction outperforms baseline MLLMs and achieves SOTA results on several other action recognition and video understanding benchmarks.

**Strengths:**

•	Benchmark Novelty: The method of using SOTA specialist models is not novel. However, the paper tackles the efficiency and effectiveness of MQA benchmarks by creating a challenging evaluation framework which successfully exposes the weaknesses of latest MLLMs model shown in Table.1.

•	Strong Performance: The proposed LLaVAction model demonstrates its significant performance gain over GPT-4o, LLAVA-OV, and LLAVA-Video on the new benchmark and also shows SOTA performance and strong generalization abilities on origin EK100 dataset and etc.

•	The paper is well-written and easy to follow, the model itself is concrete and effective: 1. Using learnable action taken with independent heads and utilize 2. Two-stage structured prediction pipeline that narrow downs the candidates choice.

**Weaknesses:**

•	Using strong external models as support: The construction relies on specialized model and other action recognition model as strong support. It might create confusion that dependence on TIM and other action recognition model will effectively improve the result rather than improvement from the model itself.

•	Possible OOD problem:
The paper does partially address my concern that they use AVION as training support vs TIM as evaluation. However, the reported 65% top-1 agreement and 45% top-5 overlap between AVION and TIM’s results on EK-100 evaluation set. The out of distribution itself will not proofed by claiming of using two different mining models. The paper might need a more robust way of concluding the OOD Distractor by possibly including a domain test on EK-100.

•	Novelty for two-stage pipeline: The paper claims the two-stage pipeline predictor as one of their contributions.  While in Section 3.3, the paper demonstrates they use a specialist model to generate Top-K candidates, it acts like a classifier for post-processing the output.

•	Specialized model confusion: In Section 4.3, the paper claims the task is zero-shot generalization. However, the specialized model is used to construct the candidate sets and also to ensure fair comparison against other model. It looks like a blur result rather than claiming as zero-shot since there is external help for inference.

**Questions:**

•	For the possible OOD problem, can you provide a clearer explanation or experiment to proof the current train/val setting remains OOD?

•	In your claim, the GPT-4o reformulation alone downgrades the MQA performance but the two-stage pipeline itself relies on GPT-4o for caption during training process. Can you elaborate the balance between benefits and harms?

•	How do results vary with K in both the miner and the two-stage pipeline? Is there a robust way to pick K beyond empirical tuning?

•	Minor notation issue: In Section 1.3, Formula 1 shows how the author formulate the MQA task. However, the author didn't definite p in the formula. Does it refer to the correct narration?

---

> ### Author Response · Authors · 2025-11-24
> **Response to Reviewer quqa (1)**
>
> We thank the reviewer for their thoughtful comments, efforts spent on our paper and the constructive feedback that will definitely make the paper better. We appreciate that they recognized our benchmark novelty and the strong performance of our paper. Below are our responses to their concerns.
>
> > 1. “Using strong external models as support: … It might create confusion that dependence on TIM and other action recognition model will effectively improve the result rather than improvement from the model itself.”
>
> - For the EPIC-KITCHENS-100-MQA benchmark, the external specialist model (TIM) is used to find hard examples and hence complicate the questions. And that requires the MLLM itself to have strong performance since GPT-4o and LLaVA models perform badly and the chance level is 20%. To make MLLMs ready for selecting the right answer from the hard questions, we proposed methods to improve their capabilities in distinguishing semantically similar actions, which is a fundamental enhancement to the MLLMs themselves.
>
> - For the zero-shot MLLM benchmarks (Table 6), LLaVAction directly takes the video and question as inputs and has no need for external specialist models. Hence, those benchmarks also require MLLM to have a strong general video understanding ability. And LLaVAction obtains consistent improvements over LLaVA-Video.
>
> - For evaluating on action recognition benchmarks that require closed-vocabulary, structured action outputs, an external model is needed in the first stage to filter out easy, irrelevant actions. However, the MLLM still needs to differentiate between the hard distracting choices in the second stage. To support that, we also evaluated LLaVA-Video-7B with the same external model on the EPIC-KITCHEN action recognition benchmark. We can see the performance of LLaVA-Video-7B is much worse, even with an external model, showing that it struggles to solve the hard distractors
>
> | Methods                          | Top1 Acc.  |
> |-----------------------|------------------|
> | LLaVA-Video-7B           | 26.49          |
> | LLaVAction-7B              | 58.3            |
>
>  > 2. “Novelty for two-stage pipeline: The paper claims the two-stage pipeline predictor as one of their contributions. While in Section 3.3, the paper demonstrates they use a specialist model to generate Top-K candidates, it acts like a classifier for post-processing the output.”
>
> - The main novelty of our proposed two-stage pipeline is that it enables the MLLM to fairly compare and outperform other SOTA action recognition methods. MLLMs directly output free texts, which makes it hard to find an exact match with the action labels in the dataset, especially when the action space is huge and fine-grained. To support that, we implement another way to achieve structured action output. Specifically, we first prefill and store the KV Cache for video+prompt+question to avoid repeated computation. After that, we append each action class to compute the text cross-entropy loss. The action class with the lowest loss is selected as the final action prediction. We test for both the zero-shot LLaVA-Video-7B and our LLaVAction models on the EPIC-KITCHEN action recognition benchmark. The results are as follows.
>
> | Methods                  | Top1 Acc.  |
> |-----------------------|------------------|
> | LLaVA-Video-7B   | 10.34         |
> | LLaVAction-7B      | 40.03         |
>
> - We can see that both LLaVA-Video-7B and our LLaVAction-7B perform much worse in this one-stage approach compared to using our two-stage inference pipeline (26.49 and 58.3).  In summary, our method sheds some light on how MLLMs can work with specialized models to mitigate the burden of reasoning over all possible candidates for tasks like action recognition. Future works can also explore adding a specialized module, thus integrating a candidate proposal network in the architecture, much like how the two-stage object detector evolved.
>
> - Finally, the hard-example mining approach that makes the two-stage pipeline possible also consistently improves the MLLM’s performance across action understanding benchmarks without the specialized models, suggesting that distinguishing semantically similar actions is a strong learning signal. This should also be considered as a novelty.

---

> > ### Author Response · Authors · 2025-11-24
> > **Response to Reviewer quqa (2)**
> >
> > > 3. “Possible OOD problem: …  The paper might need a more robust way of concluding the OOD Distractor by possibly including a domain test on EK-100.”, “For the possible OOD problem, can you provide a clearer explanation or experiment to prove the current train/val setting remains OOD?”
> >
> > - An important signature of the IID vs. OOD argument is that OOD is less vulnerable to overfitting when giving the model more chances to explore the training data. Following our OOD setting that uses Avion distractors for training and TIM distractors for testing,  we performed experiments that vary K in both training and testing, for K = 5, 10, 20, we got 74.3, 69.5, 64.1, respectively. Since test-time distractors are generated by TIM and training-time distractors are generated by AVION, we believe increasing K in training introduces overfitting, and thus it does not generalize well to TIM’s distractors, which further supports our rationale for the OOD setting. In comparison, we have observed that increasing K will generally improve our EK-100 benchmark performance (from K=5 to K=20 and saturate around K = 20) if we use TIM distractors also for training.
> >
> > - Additionally, we computed the Jensen–Shannon Divergence (JSD) between the softmax outputs of Avion and Tim across the validation set (n = 9,668). The mean JSD was 0.674 ± 0.089, with a 95% confidence interval of [0.672, 0.765]. We obtained a p-value < 0.001 and a Cohen’s d of 7.57, indicating a large and statistically significant difference between Avion and Tim. As a consequence, their generated distractors should be seen as coming from two different distributions.
> >
> > - Finally, we want to mention that our OOD is relative to our IID setting that uses TIM as distractors. Different data split and different distractors prevent our MLLM from purely overfitting on TIM’s prediction. With that, we are able to obtain consistent performance improvements on other zero-shot MLLM benchmarks.
> >
> > > 4. “ Specialized model confusion: In Section 4.3, the paper claims the task is zero-shot generalization. However, the specialized model is used to construct the candidate sets and also to ensure fair comparison against other model. It looks like a blur result rather than claiming as zero-shot since there is external help for inference.”
> >
> > - We believe you are referring to the zero-shot results in Table 5 (EPFL-Smart-Kitchen-30). There, we aimed to demonstrate the generalization of models as the second-stage model (selecting the correct action from candidates generated by the first-stage model). Therefore, zero-shot means our MLLM model is not trained on EPFL-Smart-Kitchen-30, never seeing the video and having no knowledge about the label space. We would make this clearer in the revised manuscript.
> >
> > - However, it needs to be noted that it’s still a fair comparison in the sense that AVION and LLaVA-Video are also leveraging the same VideoMAE as the external model. And our MLLM model can obtain better tail action performance (24.6) compared to the base external model (19.4). Both comparisons can show the generalization ability of our method.
> >
> > > 5. “In your claim, the GPT-4o reformulation alone downgrades the MQA performance but the two-stage pipeline itself relies on GPT-4o for caption during training process. Can you elaborate the balance between benefits and harms?”
> >
> > - In section 4.2, we explained the behavior of performance degradation to ‘catastrophic forgetting MQA’. Indeed, we used GPT-4o for free-form tasks such as captions and reasoning before prediction. We explained in section 4.2 that the GPT-4o reformulation improves the performance only when it combines with MQA tasks. We think that free-form tasks, such as caption and reasoning before prediction, will improve the MQA task and vice versa, given that catastrophic forgetting of tasks is handled.
> >
> > > 6. “How do results vary with K in both the miner and the two-stage pipeline? Is there a robust way to pick K beyond empirical tuning?”
> >
> > - In early experiments, we experimented with different K as well as different sources of distractors. We observed that if we picked Avion as the training distractor and varied the K for both training and inference, we would obtain 74.3, 69.5, and 64.1 for K = 5, 10, and 20, respectively. This suggests that picking K too large has the potential of overfitting to Avion’s distribution and since a larger K is less common in the VQA pretraining of the base MLLM (such as LLaVA-Video or LLaVA-OneVision), a larger K might hinder the model from using its pre-existing knowledge. Therefore, we picked K = 5 for LLaVAction for the best generalization capabilities.
> >
> > > 7. “Minor notation issue: In Section 1.3, Formula 1 shows how the author formulate the MQA task. However, the author didn't definite p in the formula. Does it refer to the correct narration?”
> >
> > - We apologize for the omission. p_i is the probability of picking the i-th option in the MQA as the answer. We will fix this in the newest manuscript.

---

> > > ### Author Response · Authors · 2025-11-26
> > > **Kind Reminder: Seeking Reviewer Feedback for Author/Reviewer Discussion Phase**
> > >
> > > Dear reviewer quqa,
> > >
> > > We hope this message finds you well. We would like to express our gratitude for the time and effort that you have dedicated to the review process of our submission.
> > >
> > > Furthermore, we are writing to kindly remind you that the rebuttal period is coming to a close. Your feedback and evaluation are crucial to the progress of our work, and we value your expertise and insights immensely.
> > >
> > > During the rebuttal, we hope we have addressed all the queries and concerns you raised regarding our submission. Each point has been carefully considered, and we have provided detailed responses.
> > >
> > > We are eager to engage in further discussions with you on the responses and any other aspects of our study. Your feedback plays a pivotal role in shaping the direction of our research, and we are keen to hear your thoughts and suggestions.
> > >
> > > Thank you for your consideration.
> > >
> > > Best regards,
> > >
> > > Authors of submission #9309

---

### Official Review · Reviewer_HUeB · 2025-11-01

**Soundness:** 3
**Presentation:** 2
**Contribution:** 2
**Rating:** 6
**Confidence:** 4

**Summary:**

The paper introduces LLaVAction, a novel framework designed to evaluate and enhance the fine-grained action understanding capabilities of multi-modal large language models (MLLMs). The authors first reformulate the EPIC-KITCHENS-100 dataset into a video multiple-choice QA benchmark (EPIC-KITCHENS-100-MQA) using hard example mining with state-of-the-art specialist models such as TIM to generate challenging distractors. This benchmark exposes the current limitations of existing MLLMs (e.g., GPT-4o) in fine-grained action discrimination.
To improve MLLM performance, the authors curate a supervised fine-tuning dataset encompassing tasks like hard action recognition, temporal detection, captioning, and free-form QA. The proposed LLaVAction model introduces an action token to strengthen visual information utilization and a two-stage structured prediction pipeline for fair comparison with specialized models.
Empirical results show significant improvements—over 21-point accuracy gains on EPIC-KITCHENS-100-MQA compared to GPT-4o--and strong generalization across multiple benchmarks, suggesting that LLaVAction effectively enhances MLLMs’ action understanding.

**Strengths:**

Comprehensive evaluation across multiple datasets, benchmarks, and modalities.


Innovative benchmark creation using model-based hard example mining—a principled and efficient alternative to human-generated distractors.


Clear empirical improvement over strong baselines (e.g., GPT-4o, LLaVA-Video).


Thoughtful model design, integrating both architectural and data-centric enhancements.


Strong generalization across domains, including unseen datasets like EPFL-Smart-Kitchen-30 and Meccano.


Interpretability analysis (Section 4.5) provides convincing evidence of improved visual-text alignment.

**Weaknesses:**

While comprehensive, the paper’s novelty leans more on system-level integration than on introducing fundamentally new model architectures.


The action token mechanism—though effective—could benefit from deeper theoretical or ablation-based exploration (e.g., why it captures action semantics more effectively than other token-aggregation methods).


The reformulation pipeline is heavily reliant on existing specialist models (e.g., TIM, AVION), which could limit generalizability to non-egocentric domains.


The benchmark’s dependence on EPIC-KITCHENS-100 might introduce dataset bias, and expanding evaluation to non-cooking activities would further strengthen claims of generalization.

**Questions:**

Could the authors discuss how LLaVAction’s two-stage pipeline scales to open-vocabulary or long-horizon tasks beyond EPIC-KITCHENS?


Are there any failure cases where hard distractors cause misalignment between textual and visual cues?


How sensitive is LLaVAction to the choice of the base MLLM (e.g., Qwen2 backbone)?


Would the authors consider releasing the adversarially generated distractor datasets for community benchmarking?

In related works, the authors should talk about how their method compares to other methods like InsTALL [Nguyen et al., 2025 arXiv].

How do the results look with more traditional methods like ResNet3D, InceptionNet-3D (I3D), etc.?

---

> ### Comment · Reviewer_HUeB · 2025-11-20
>
> Please let me know if there are any questions about the review. Thanks.

---

> ### Author Response · Authors · 2025-11-24
> **Response to Reviewer HUeB (1)**
>
> We thank the reviewer for their thoughtful comments and the time and effort spent reviewing our paper. We appreciate that they recognized the innovation of our hard example mining and the corresponding benchmark, as well as model design, performance, generalizability and interpretability. Below are our responses regarding the concerns.
>
> > 1. “Novelty leans more on system-level integration than on introducing fundamentally new model architectures.”
>
> - Our key aim is to evaluate and improve the action understanding ability of MLLM instead of introducing a completely new MLLM architecture. Our novelty lies in action understanding related data reformulation and model designs that could be used for any foundational MLLM, including the hard example mining and learnable action token. And that is why we mainly compare with LLaVA-Video and show improvements over LLaVA-Video on both action recognition and MLLM benchmarks. We will make this clearer in the revised manuscript.
>
> > 2. “ Action token mechanism—though effective—could benefit from deeper theoretical or ablation-based exploration”
>
> - For theoretical explanation, the action token is learnable and we use the action classification loss to supervise its learning, which is analogically similar to the CLS token in the VIT model. Previous work by Wang et al. has shown the ability of VIT’s CLS token in grasping image content [1]. Therefore, by putting our action token before the question token and after the visual tokens, our action token could also see and encode action-related video information, and hence benefit the question-answering task.
>
> - For ablation-based exploration, our additive ablations (Table 3) validate its contribution to the model performance and attention-based analysis (Section 4.5) validates its effectiveness in an interpretable way. To further support our action token design, as recommended by the reviewer, we implement several other token-aggregation ways. Specifically, our action token is one learnable token added between the visual tokens and text tokens and is supervised in the last layer with the action classification loss. We hence implement three variants: 1) adding three action tokens and supervising them with verb, noun, and action separately in the last layer; 2) adding one action token and supervising it in the first layer of the MLLM; adding one action token and supervising it across all MLLM layers. The results are as follows on our benchmark. We can see our action design performs the best while keeping simplicity.
>
> | Token designs                | Top1 Acc.  |
> |------------------------------|---------------|
> | 1). 3 tokens, last layer    | 68.8           |
> | 2). 1 token, first layer    | 66.2           |
> | 3). 1 token, all layers     | 31.7            |
> | Ours. 1 token, last layer | 69.1            |
>
> > 3. “The reformulation pipeline is heavily reliant on existing specialist models (e.g., TIM, AVION), which could limit generalizability to non-egocentric domains.”
>
> - Firstly, our action-related MLLM data reformulation pipeline contains several reformulation ways. Ways such as temporal detection, temporal order learning and direct prediction do not require specialist models. Our adversarial distractors do need to rely on specialist models. However, compared to human-based reformulation and GPT-based reformulation, the time and expense costs to obtain hard and fine-grained data samples are much lower. Meanwhile, our reformulation pipeline has no constraint in applying to non-egocentric domains or choosing different specialist models. Please see the 4th response about the animal experiments.

---

> > ### Author Response · Authors · 2025-11-24
> > **Response to Reviewer HUeB (2)**
> >
> > > 4. “Generalizability to non-egocentric domains”, “Expanding evaluation to non-cooking activities”, “The choice of the base MLLM (e.g., Qwen2 backbone)”
> >
> > - For non-cooking activities, we have tested our method on the Meccano dataset, which is targeted for tool assembling. LLaVAction obtained 51.7 top-1 accuracy, beating SlowFast’s 42.8 (Sec. 4.3). For the choice of base MLLM, we have adapted LLaVA-Video-7B, LLaVA-OV-7B and LLaVA-OV-0.5B and all of them show much better performances (Table 2).
> >
> > - In general, our action understanding-related data reformulation and model designs are not limited to any data domains or base MLLMs (e.g., Qwen2). To further support that, we tested our method on a very different domain, animal fine-grained behavior understanding. Specifically, we tested on the Animal Kingdom dataset [2], which has 140 fine-grained actions and at most 12 actions can happen at the same time (multi-classification task). We adapted both LLaVA-Video-7B (Qwen2 backbone) and InternVL3-8B (InternLM2 backbone) and used VideoMAE to generate hard examples. The results are as follows, reported in Jaccard accuracy. We can see LLaVAction can obtain much better performance compared to zero-shot and finetuned with random choice using different base MLLMs. And both base MLLMs can further beat the baseline VideoMAE.
> >
> > | Methods                | Jaccard Acc.  |
> > |-----------------------|------------------|
> > | VideoMAE           | 53.13
> > |**LLaVA-Video-7B**|
> > | Zero shot              | 30.49             |
> > | Random choice    | 46.73             |
> > | Ours         | 61.00             |
> > |**InternVL3-8B**|
> > | Zero shot             | 28.08             |
> > | Random choice   | 43.86             |
> > | Ours        | 58.73             |
> >
> > > 5. “LLaVAction’s two-stage pipeline scales to open-vocabulary or long-horizon tasks beyond EPIC-KITCHENS”
> >
> > - Our two-stage pipeline is only needed for those datasets/applications that require structured actions. For the open-vocabulary or long-horizon tasks. LLaVAction can directly take the video and questions and give the open-ended answer. We have tested LLaVAction on very different MLLM video benchmarks (Table 6) that require different action understanding abilities. Specifically, the video caption and open-ended QA benchmarks (e.g., VDC and CVRR) can show LLaVAction’s open-vocabulary ability. And benchmarks that contain very long videos (e.g., VideoMME and LongVideoBench) can show LLaVAction’s long-horizon ability. LLaVAction can show consistent improvement over LLaVA-Video.
> >
> > > 6. “Failure cases where hard distractors cause misalignment between textual and visual cues”
> >
> > - During training and benchmark construction, we always include GT action into the choices to avoid misalignment. During the evaluation of the action recognition benchmark. GT action could not be included in the choices and LLaVAction cannot predict correctly in this situation. But this can make a fair comparison with existing action recognition methods and LLaVAction outperforms them.
> >
> > > 7. “Releasing the adversarially generated distractor datasets for community benchmarking”
> >
> > - Yes, we would release the codebase, all the model weights, the whole benchmark and the re-formulated training data upon acceptance.

---

> > > ### Author Response · Authors · 2025-11-24
> > > **Response to Reviewer HUeB (3)**
> > >
> > > > 8. Comparison to InsTALL [Nguyen et al., 2025 arXiv]
> > >
> > > - Thanks for pointing out the related literature and we will include it in our related work. InsTALL is a VideoLLM that also enhances MLLM’s capability on action understanding. There, they focus more on procedure/step-related actions that take context into account. In particular, they show that including Procedure Graph generally improves those tasks and the authors' hypothesis was that it relieves the burden of MLLM’s reasoning on procedural actions, which MLLMs are not yet good at as well.
> > >
> > > - Similarities: Both LLaVAction and InsTALL recognize that ‘Direct prediction’ (Equation 1-4 in InsTALL) on action recognition helps. And both works include ‘context’ into action learning as actions are naturally continuous, though we use simple yet effective ways of including previous prior actions during training and inference and they mostly focus on including procedure graphs in the inference, similar to RAG approaches.
> > >
> > > - Main differences:  InsTALL mainly focuses on improving procedural action planning/prediction and barely focuses on differentiating fine-grained actions. In comparison,  LLaVAction focuses more on evaluating and improving MLLMs’ capabilities on telling apart fine-grained actions, which is both an under-explored weakness of MLLMs and good learning signals to improve MLLMs’ action understanding capabilities (demonstrated by generalization on video understanding benchmarks and other action recognition datasets).
> > >
> > > > 9. “Results for more traditional methods like ResNet3D, InceptionNet-3D (I3D)”
> > >
> > > - As far as we know, ResNet3D has not been applied to EPIC-Kitchen-100. IPL[3] utilized the I3D model and obtained 41.0 top 1 accuracy. All of the traditional methods lag much behind our performance.
> > >
> > > [1]. Wang, Yuxuan, et al. "VideoLLaMB: Long Streaming Video Understanding with Recurrent Memory Bridges." Proceedings of the IEEE/CVF International Conference on Computer Vision. 2025.
> > >
> > > [2]. Ng, Xun Long, et al. "Animal kingdom: A large and diverse dataset for animal behavior understanding." Proceedings of the IEEE/CVF conference on computer vision and pattern recognition. 2022.
> > >
> > > [3]. Wang, Xiaohan, et al. "Interactive prototype learning for egocentric action recognition." Proceedings of the IEEE/CVF International Conference on Computer Vision. 2021.

---

> > > > ### Author Response · Authors · 2025-11-26
> > > > **Kind Reminder: Seeking Reviewer Feedback for Author/Reviewer Discussion Phase**
> > > >
> > > > Dear reviewer HUeB,
> > > >
> > > > We hope this message finds you well. We would like to express our gratitude for the time and effort that you have dedicated to the review process of our submission.
> > > >
> > > > Furthermore, we are writing to kindly remind you that the rebuttal period is coming to a close. Your feedback and evaluation are crucial to the progress of our work, and we value your expertise and insights immensely.
> > > >
> > > > During the rebuttal, we hope we have addressed all the queries and concerns you raised regarding our submission. Each point has been carefully considered, and we have provided detailed responses.
> > > >
> > > > We are eager to engage in further discussions with you on the responses and any other aspects of our study. Your feedback plays a pivotal role in shaping the direction of our research, and we are keen to hear your thoughts and suggestions.
> > > >
> > > > Thank you for your consideration.
> > > >
> > > > Best regards,
> > > >
> > > > Authors of submission #9309

---

> > > > > ### Comment · Reviewer_HUeB · 2025-11-27
> > > > >
> > > > > Thank you for the detailed rebuttal response. i look forward to seeing these in the main text/appendix of the paper for further understanding of the work to the end user.

---

> > > > > > ### Author Response · Authors · 2025-11-27
> > > > > > **Revision of the Manuscript**
> > > > > >
> > > > > > Dear reviewer HUeB,
> > > > > >
> > > > > > Thank you very much for your reply. We have updated the manuscript with the new experiments and added more clarifications based on reviewers’ questions (with revisions marked in red). Please let us know if there are other things to be added to the manuscript or if you have any additional comments regarding our paper.
> > > > > >
> > > > > > Best regards,
> > > > > >
> > > > > > Authors of submission #9309

---

### Author Response · Authors · 2025-11-27
**Revision of the Manuscript**

Dear Reviewers,

Thank you once again for your valuable feedback.

We have revised the manuscript in accordance with our first-round discussion, and we hope that our responses and the updated version address all the questions and concerns you raised. We would be happy to continue the discussion if you have any additional comments.

Below, we summarize the changes made to the updated PDF document (with revisions marked in red).

### **Abstract**

- `HUeB, V9cL`: adding open-source statement.

### **Section 1 Introduction**

- `HUeB`: adding clarification about the model designs.
- `V9cL`: adding clarification about the action understanding ability.

### **Section 2 Related works**

- `HUeB, V9cL`: adding related works for action understanding MLLMs (InsTALL, HAIC, MotionLLM).

### **Section 3 Methods**

- `quqa`: fixing the notation issue in Section 3.1.
- `BpZo`: adding more details about choice generation in Section 3.1.
- `quqa`: adding clarification about the OOD setting in Section 3.2.
- `HUeB`: adding theoretical explanation for action token in Section 3.3.
- `HUeB`: adding clarification about the two-stage approach in Section 3.3.
- `BpZo`: adding clarification about action token’s usage during training and inference in Section 3.3.

### **Section 4 Experiments**

- `HUeB`: adding comparison with more traditional method (I3D) in Table 4.
- `quqa`: adding clarification about the  zero-shot generalization experiment in Section 4.3 and Table 5.
- `HUeB, BpZo`: adding Animal Kingdom experiment in Section 4.3 and Table 6.

### **Appendix**

- `quqa`: adding more supports about the OOD setting in Appendix D.
- `HUeB`: adding action token ablation study in Appendix E.1.
- `BpZo, quqa`: adding structured action prediction ablations in Appendix E.3.

Best regards,

Authors of submission #9309

---

### Author Response · Authors · 2025-12-01
**Summary for the rebuttal by Authors**

Firstly, we’d like to thank the reviewers for acknowledging the strengths of our work.

**Method novelty**

- `HUeB`: *“Innovative benchmark creation using model-based hard example mining”, “Thoughtful model design, integrating both architectural and data-centric enhancements”.*

- `quqa`: *“The paper tackles the efficiency and effectiveness of MQA benchmarks”, “A challenging evaluation framework which successfully exposes the weaknesses of latest MLLMs model”.*

- `V9cL`: *“The research topic is highly practical and meaningful … could benefit a wide range of real-world applications”, “A new, challenging benchmark … to contribute valuable insights to research within the relevant domain”.*

- `BpZo`: *“Comprehensive training data mixture .. to teach the model action boundaries and natural continuity”.*

**Experiments & Results**

- `HUeB`: *“Comprehensive evaluation across multiple datasets, benchmarks, and modalities”, “Clear empirical improvement over strong baselines”, “Strong generalization across domains”, “Convincing evidence of improved visual-text alignment”.*

- `quqa`: *”Significant performance gain over GPT-4o, LLAVA-OV, and LLAVA-Video on the new benchmark”, “SOTA performance and strong generalization abilities on origin EK100 dataset and etc”.*

- `V9cL`: *“Strong performance in action recognition tasks”, “Improvements across more generic benchmark datasets … showcasing the potential value and versatility”.*

- `BpZo`: *“Improved performance over baselines like GPT-4o”, “Training data mixture strategy improves action understanding ability of MLLMs”.*

**Presentation**

- `quqa`: *“The paper is well-written and easy to follow”.*

- `V9cL`: *“The paper is well-written … in a clear and systematic manner”.*

Secondly, we thank the reviewers for providing constructive feedback, which has been carefully considered. We responded individually, and updated the manuscript accordingly. In summary, the major points are as follows.

- We verified the generalization ability of LLavaction. Firstly, we showed that it also applies to other domains (animal behavioral analysis). Secondly,  we showed consistent results when using a different base LLM (InternLM2). We also clarified that LLavaction achieved better general action understanding as a standalone model.

- We ablated the action token design and showed its effectiveness. We also improved the presentation of this part.

- We showed the advantage of the two-stage pipeline in applying MLLM for structured action prediction by comparing it with a variant proposed by Reviewer `BpZo`. Our two-stage pipeline is 190 times faster and achieves significant improvements in both zero-shot (10.3→26.5) and finetuned (40.0→58.3) settings.

During the rebuttal, we have addressed all the queries and concerns since no further concerns have been raised and two reviewers have recognized our detailed rebuttal response.

Thanks for all your time. Your feedback plays a pivotal role in shaping the direction of our research.

---

### Meta-Review · Area_Chair_Nvfe · 2026-01-13

**Summary:**

This paper proposes an EPIC-KITCHENS based benchmark containing difficult action-related Q&A, as well as supervised finetuning dataset containing hard action recognition, temporal detection, captioning, and free-form Q&A data (trained with an action token mechanism) yielding LLaVAction. Reviewers raised a number of important concerns shared across reviewers, especially regarding the novelty of the method (as this paper is largely a benchmark and then SFT dataset, with minor changes through the action token mechanism which as BpZo points out reduces generality), requirement of the two-stage method compared to simpler one-stage ones, the reliance of specialist models preventing generalization, potential biases in the dataset (i.e. action recognition vs. understanding, cooking activities), and claims in terms of the OOD setting.

**Reviewer Concerns:**

Many of the concerns, such as generality to other domains beyond cooking, etc. were well-addressed, including with new experiments on animal behavior understanding tasks. The authors showed a few more experiments demonstrating the need for the two-stage method as well, and some aspects of generalizability (to non-action domains) were addressed through the Meccano dataset as well as existing tables showing performance across a variety of diverse datasets.

Some aspects of novelty/contribution, however, were not as well-addressed. For example, the statement "The main novelty of our proposed two-stage pipeline is that it enables the MLLM to fairly compare and outperform other SOTA action recognition methods" does not really describe novelty and indeed the method is quite simple and akin to many other small architectural variations.

**Reviewer Scores:**

The positive reviewers (HUeB and V9cL) maintain their score, as both expressed satisfaction with the rebuttal and one explicitly mentioned retaining their rating. Reviewer quqa is not as likely to increase their score, in my assessment, as questions about novelty/contribution (as mentioned above) are not as convincing. Reviewer BpZo had a different set of questions regarding especially the action token, which were answered for the most part.

Overall, this paper is therefore borderline leaning toward acceptance. While the novelty of the method is not strong, the dataset creation methodology (even though it requires strong specialist models for each domain) and comprehensive experimentation/analysis may be interesting to the MLLM and action understanding communities.

---

### Decision · Program_Chairs · 2026-01-26

Accept (Poster)